# Artificial fingerprints engraved through block-copolymers as nanoscale physical unclonable functions for authentication and identification

Irdi Murataj [1,4], Chiara Magosso [1,2,4], Stefano Carignano [3], Matteo Fretto[1], Federico Ferrarese Lupi [1] ✉ & Gianluca Milano [1] ✉

Besides causing financial losses and damage to the brand's reputation, counterfeiting can threaten the health system and global security. In this context, physical unclonable functions (PUFs) have been proposed to overcome limitations of current anti-counterfeiting technologies. Here, we report on artificial fingerprints that can be directly engraved on a wide range of substrates through self-assembled block-copolymer templating as nanoscale PUFs for secure authentication and identification. Results show that morphological features can be exploited to encode fingerprint-like nanopatterns in binary code matrices representing a unique bit stream of information characterized by high uniqueness and entropy. A strategy based on computer vision concepts for authentication/identification in real-world scenarios is reported. Long-term reliable operation and robust authentication/identification against thermal treatment at cryogenic and high temperatures of the PUF have been demonstrated. These results pave the way for the realization of PUFs embracing the inherent stochasticity of self-assembled materials at the nanoscale.

Counterfeiting represents a growing problem for society that has not only enormous economic implications, from manufacturing of goods to specialized technologies but represents a threat for the entire security and health systems[1]. In the current era of Internet of Things (IoT), widely exploited anti-counterfeiting techniques based on tags such as graphical barcodes, watermarks, and holograms realized through a deterministic process endow low complexity and high predictability, allowing easy forgery and cloning by counterfeiters[2]. Also, widely exploited software-based authentication/identification systems endow intrinsic vulnerability due to electromagnetic interference and cyber-attacks[3]. To overcome main limitations of current anti-counterfeiting technologies, hardware encryption systems based on physical unclonable functions (PUFs) have attracted a growing interest[4,5]. These are physical one-way functions that enable the generation of a secret key based on the inherent physical characteristics of a physical system/device, where the uniqueness of the key relies on uncontrollable variabilities and high randomness of the physical system even when realized through the same fabrication process[6]. In this framework, new perspectives for the realization of PUFs can be enabled by exploiting the true essence of nanotechnology where the fabrication of materials and systems at the nanoscale is often hindered by unpredictable disturbances and fluctuations[7]. This is because at this scale the influence of thermal and statistical fluctuations cannot be avoided, resulting in output material structures and properties that are not simply determined by known inputs. Leveraging the inherent stochasticity of nanotechnologies, a wide range of physical unclonable

[1]Advanced Materials Metrology and Life Sciences Division, INRiM (Istituto Nazionale di Ricerca Metrologica), Turin, Italy. [2]Department of Electronics and Telecommunications, Politecnico di Torino, Turin, Italy. [3]Barcelona Supercomputing Center, Barcelona, Spain. [4]These authors contributed equally: Irdi Murataj, Chiara Magosso. ✉e-mail: f.ferrareselupi@inrim.it; g.milano@inrim.it

functions have been demonstrated by employing different taggants and detection methods. This includes random fractal structures[8], fluorescent, photoluminescent and chromic materials[9–12], spontaneously formed core-cell nanoparticles[13], 3D printed and nanoimprinted materials[14,15], plasmonic nanomaterials[16–18], graphene-based devices[19], as well as metallic nanopatterns replicated from molecular self-assembly[20,21]. These techniques mainly imply the realization of unique nanopatterned devices as anti-counterfeiting tags that can be affixed to products requiring protection, but that can potentially be removed or detached through appropriate chemical/physical treatments.

Here, we report on artificial nano fingerprints that can be directly engraved in a wide range of target materials as disorder based PUFs for secure authentication and identification, characterized by long-term reliable operation and high thermal stability. Besides proposing a strategy for user-independent extraction of genuine fingerprint-like nanopatterns without artifacts from micrographs, we show that by exploiting local features of nanopatterns it is possible to encode nano fingerprints in unique binary code matrices endowing high bit uniformity, high uniqueness, and high entropy. Moreover, we propose and test a computer vision-based strategy for robust authentication/identification through artificial fingerprints in real-world scenarios, demonstrating image-based PUFs as anti-counterfeit identifiers that utilizes the intrinsic unpredictability of self-assembling materials as a source of randomness. Furthermore, we demonstrate a high stability of the PUF against aging and thermal stability, an aspect often neglected, providing evidence of long-term reliable operation and robust authentication/identification in the thermal range from −196 °C to 200 °C.

## Results

### Nanoscale artificial fingerprints engraved on target substrates

Since ancient Babylon, fingerprints have been exploited as unique identifiers to secure commercial transactions, as testified by clay seals attached to their business documents[22]. In the era of Internet of Things, fingerprint matching is largely deemed to be a secure system for both authentication (i.e., matching a person's biometric template) and identification (i.e., determining the identity of a person), as schematized in Fig. 1a. Here, the validation and verification of a person's identity are based on unique physical characteristics—biometrics—of their fingerprints[23]. The security of this process is guaranteed by the fact that, although two or more fingerprints may share the same global features, there are currently no known pairs of fingerprints with identical pattern of local features, also called minutiae points[24]. Besides the individuality of the fingerprint, the premises for fingerprint authentication/identification rely on the pattern persistence, meaning that the characteristics of fingerprints must be stable over time[25]. Similarly, artificial fingerprints can be exploited as PUFs for the authentication and/or identification of a product in the supply chain (the workflow of PUF generation, database initialization, and authentication is schematized in Fig. 1b). Inspired by fingerprint biometrics, we developed a process to directly transfer a nano fingerprint pattern into a wide range of target substrates, as schematized in Fig. 1c. This process is based on i) molecular self-assembly of block copolymers (BCPs) on the target substrate, ii) selective removal of one of the BCP phase, and iii) pattern transferring to the target substrate. Molecular self-assembly relies on phase separation of chemically distinct and thermodynamically incompatible polymeric components of BCPs that occurs randomly under thermal fluctuations, resulting, under proper conditions, in lamellar fingerprint-like patterns (Supplementary Note 1). After achieving the self-assembled pattern, one of the two polymers comprising the BCP can be selectively removed, resulting in the formation of a disordered nanolithographic mask. Figure 1d reports an example of self-assembled lamellar BCP showing fingerprint-like features (fabrication details in Methods). Notably,

these patterns are characterized by the presence of spatially distributed local features (defect points) closely resembling minutiae points of human fingerprints. In case of lamellar patterns, the defect taxonomy of both positive and negative phases includes terminal points, 3-way junctions and dots (Fig. 1e). While the overall morphology of BCP patterns (i.e., the lamellar shape with a given periodicity and long-range ordering) is determined by thermodynamic principles and can be tailored through the appropriate choice of BCP molecular weights and processing conditions[26,27], the finer nanopattern characteristics exhibit elements of randomness. This randomness is due to local fluctuations, kinetic factors, and defects that naturally arise during the self-assembly process[26]. Furthermore, a polydispersity index (PDI) different from 1 contributes to irregularities, imperfections, and the random formation of defects in the pattern[28]. In this context, it is worth noting that recent advancements in the synthesis of BCP with covariant properties allow for on-demand tuning of morphological characteristics (e.g., periodicity, line-edge roughness), long-range order, and defectiveness[29].

After achieving the self-assembled pattern, one of the two polymers comprising the BCP can be selectively removed, resulting in the formation of a disordered nanolithographic mask. The polymeric fingerprint-like pattern can be then transferred to the target substrate through selective reactive ion etching (RIE) or wet chemical etching, depending on the material of the target substrate. Examples of patterns successfully transferred to diamond, Si, quartz, and $SiO_2$ target substrates are reported in Fig. 1f–i (details of pattern transfer on different substrates in Methods). Additionally, patterns have been demonstrated to be transferable on metallic (Cu, Cr, Co, W, CoCrPt, Permalloy)[30–35], dielectric ($Si_3N_4$, $TiO_2$)[36,37], semiconductor (ITO, graphene)[30,35] and flexible substrates[38,39]. Notably, the possibility of creating fingerprint patterns conformal to 3D substrates made from commercial optical resins or graphene liquid crystalline fibers has also been demonstrated[40]. Even if substrate roughness can affect the self-assembly process[41,42], several studies demonstrated the possibility of realizing self-assembled BCP nanostructures even on rough and irregular surface[38,40,43]. As an example, we tested BCP self-assembly on the rough surface of a rolled metal foil and on a clock hand of nickel-plated brass, as reported in Supplementary Figs. S1 and S2, respectively. The ability to conduct the self-assembly process of BCP on various materials and substrates opens the potential for directly engraving artificial fingerprints at the nanoscale onto a wide range of devices and products, including screens and microchips (as a proof-of-concept on a real object, we demonstrate the feasibility of the engraving approach on an uneven and rough glass lens as detailed in Supplementary Figs. S3 and S4). Note that a successful engraving process relies on the optimization of the pattern transfer process depending on both the target substrate material and roughness.

### Extraction of pattern morphology

As for human fingerprints, a critical step for exploiting nanoscale fingerprints as PUFs is represented by the automatic extraction of the fingerprint pattern including minutiae (defects) from input images. This process is crucial and may heavily rely on the quality of the input fingerprint micrographs. Since ridge structures in poor-quality fingerprint images can be not well defined, the pattern can be not correctly detected, and spurious defects can be created while genuine defects can be ignored. To increase the robustness of pattern extraction with respect to the quality of input fingerprint images, we adapted a fingerprint enhancement algorithm able to adaptively improve the clarity of ridge and valley structures based on estimated local ridge orientation and frequency[44] (Fig. 2a, details in Methods). This allows to obtain a binarized genuine fingerprint pattern that can be exploited to investigate BCP morphological parameters including defect localization (defect maps), line period, and correlation length (correlation maps)[45] through an automated analysis (details in

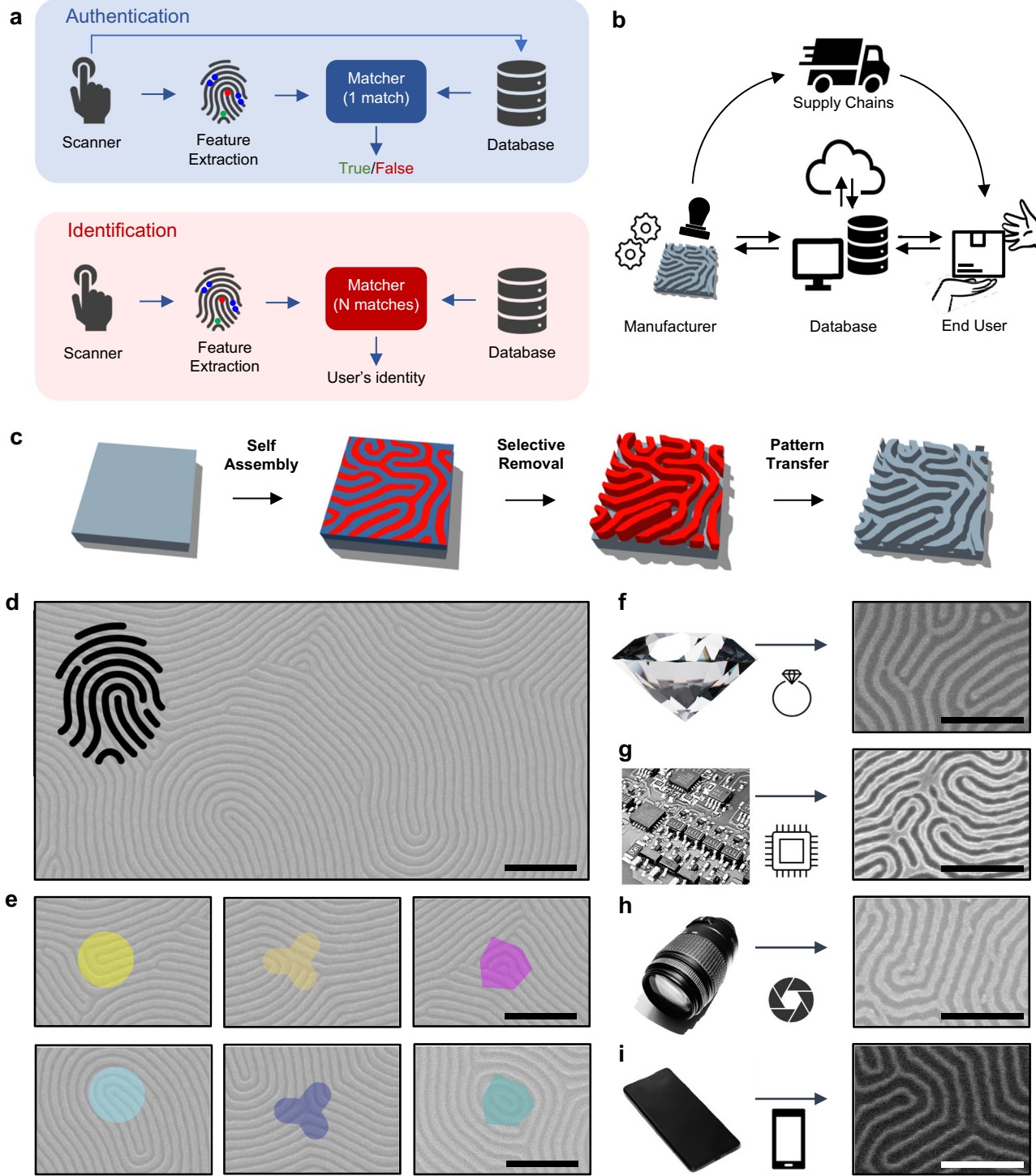

**Fig. 1 | Artificial fingerprints as physical unclonable functions for authentication and identification. a** Graphical workflow of authentication and identification processes through fingerprints. **b** Product's identification/authentication through artificial fingerprints. The artificial fingerprint can be physically transferred on the surface of the product by the manufacturer right after its production and subsequently registered and stored in the database. The product is then delivered to the buyer and the identification/authentication of the product can be performed directly by the end user by comparing the artificial fingerprint with fingerprints stored in the database. **c** Schematics of fabrication procedure by BCP self-assembly and pattern transfer for the realization of artificial nano fingerprints. **d** SEM images of self-assembled lamellar BCPs in typical fingerprint pattern-like and nanoscale defects on Si substrate. **e** Typical defects of lamellar BCPs: positive and negative phase terminal points (yellow and light blue circles, respectively), positive and negative phase 3-way junctions (orange and blue triangles, respectively), positive and negative phase dots (violet and light green squared off dots, respectively). Pattern transfer of artificial nanofingerprints on diverse surfaces, including (**f**) diamond, (**g**) Si, (**h**) quartz, and (**i**) $SiO_2$ (object images are to be considered for demonstration purposes of the items in which the engraving process can be applied to). All scale bars are set to 200 nm.

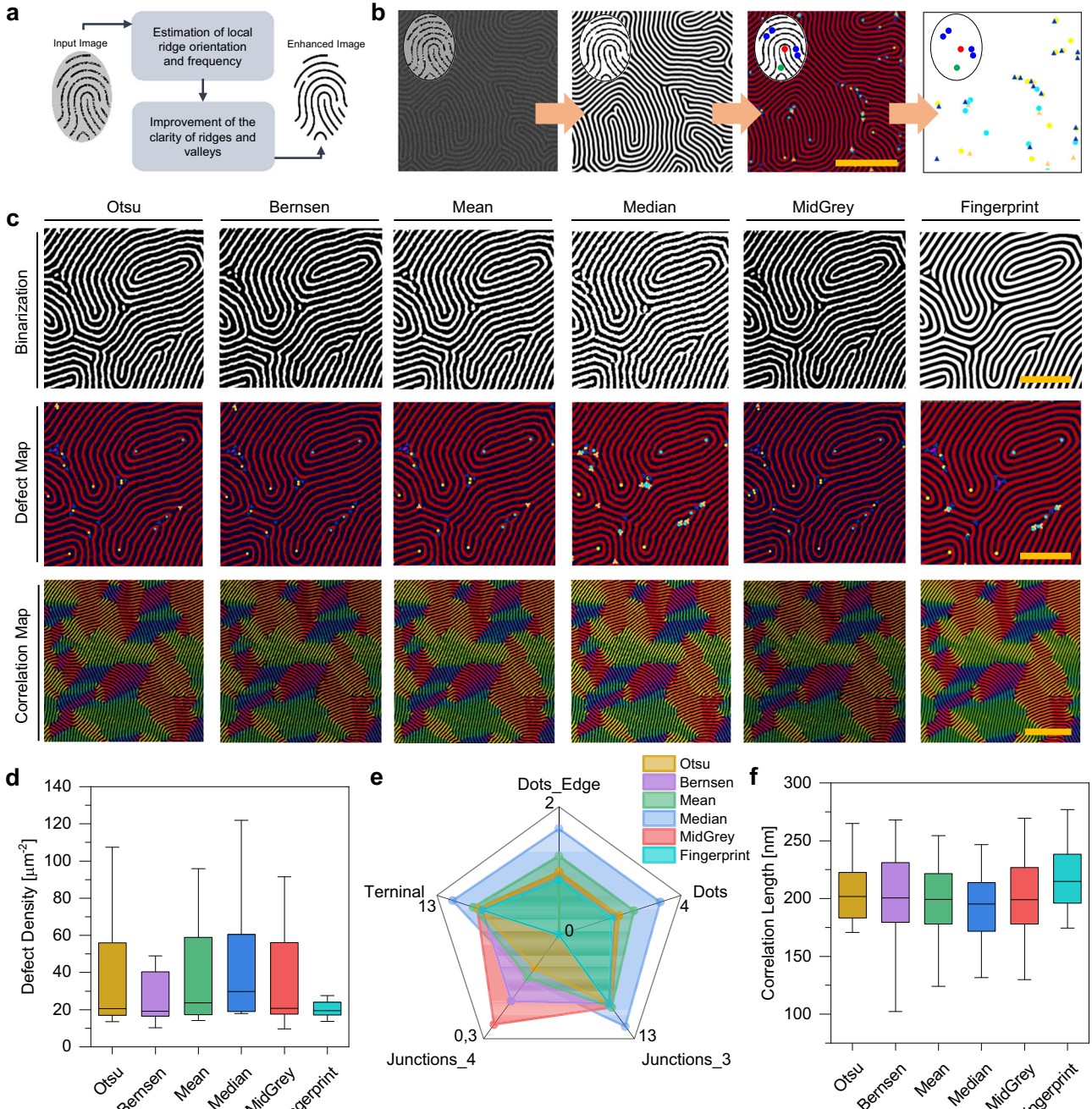

**Fig. 2 | Extraction of the artificial fingerprint nanopattern from micrographs.**
**a** Flowchart of the fingerprint enhancement algorithm exploited to extract a binarized artificial fingerprint nanopattern, based on the estimation of ridge orientation and frequency followed by filtering. **b** Example of the flow for automated extraction of morphological parameters of the nanopattern including SEM image binarization through fingerprint enhancement and creation of defect maps. Scale bar is set to 500 nm. **c** Influence of the binarization method on the extraction of the artificial fingerprint nanopattern and corresponding morphological features. Binarized nanopatterns, defect maps, and correlation maps obtained from binarized nanopatterns with conventional Otsu, Bernsen, Mean, Median, and MidGrey auto local thresholding methods are compared with the defect map and correlation map obtained from fingerprint enhancement binarization (white phase represents ridge lamellar structures, black phase represents valleys). Scale bars are set to 200 nm for Binarized nanopatterns and defect maps. Scale bar is set to 500 nm for correlation maps. **d** Defect density, (**e**) radar chart showing the density of defect types (number of defects for μm²), and (**f**) correlation length analyses of nanopatterns obtained through different binarization techniques. Data reported in panels (**d–f**) were obtained from the analysis of ~30 nanopattern micrographs (image sizes from $1.26 \times 1.48\,\mu m^2$ to $5.41 \times 6.28\,\mu m^2$). Data in (**d**, **e**) refer to defects in both positive and negative phases. **d**, **f** The boxes represent the 25th and 75th percentiles, midlines represent median values, and whiskers the 1.5 IQR (inter-quartile range).

Methods). An example of the process flow, from SEM image to fingerprint pattern enhancement and defect localization, is reported in Fig. 2b. A comparison of a binarized pattern obtained by fingerprint enhancement algorithm and other common binarization thresholding techniques on the same micrograph is reported in Fig. 2c (raw micrograph in Supplementary Fig. S5). Here, the white phase of binarized images represents ridge lamellae structures, while the black phase represents valleys. Fingerprint pattern enhancement

outperforms traditional thresholding techniques, allowing to obtain a binarized map with reduced noise and with a reduced number of artifacts (additional details in Supplementary Fig. S6). Notably, whereas the effectiveness of traditional thresholding techniques relies on the image quality, contrast, and brightness, fingerprint enhancement enables the automatized extraction of patterns even in case of poor quality and low contrast images (example in Supplementary Fig. S7). Moreover, it eliminates the need for image-dependent adjustment of binarization parameters, thereby avoiding any potential bias from the user. As depicted in defect maps (Fig. 2c), the reduced number of artifacts in images processed with fingerprint enhancement results in a limited number of spurious defects. Also, this approach enables the generation of correlation maps with a decreased noise level (Fig. 2d). A statistical analysis, performed by considering SEM micrographs acquired on different areas of the same patterned sample, allows a quantitative evaluation on the effect of the binarization extraction of defects and correlation length of the structures. Figure 2d reports the defect density obtained by analyzing images binarized with the different techniques. As evident, fingerprint enhancement results in a lower mean defect density, quantitatively showing the possibility of reducing the counting of spurious defects through this binarization technique. Furthermore, larger defect distributions obtained by conventional binarization techniques are due to overestimation of defects in images not properly binarized through these techniques (details in Supplementary Fig. S8). More in detail, the radar chart in Fig. 2e shows that conventional binarization techniques result in a higher overestimation of certain types of defects. Since defect density is inversely related to the dimensions of lamellae orientation domains[45], fingerprint-enhanced images result in a higher mean value of correlation length (Fig. 2f). All these observations show that binarization through the fingerprint enhancement algorithm allows to retrieve a genuine fingerprint pattern that, besides reducing artifacts, is user independent and can automatically extract relevant information even from poor-quality images and images acquired with different contrast/brightness conditions. These are fundamental aspects for correct authentication and/or identification of PUFs based on BCP nanopatterned substrates in real-world scenarios (example of binarization through fingerprint enhancement of a nanopattern engraved on a glass lens in Supplementary Fig. S9).

**Encoding artificial fingerprints in binary code matrices**

These artificial fingerprints can be considered as image-based PUFs, a class of PUF based on random visual features of the physical object[46], that can be exploited as anti-counterfeit tags and/or identifiers. Specifically, the nanopattern represents the unique response $r$ when the surface is scanned with a focused beam of electrons representing the input challenge $c$. In this context, the uniqueness of the PUF response $r = f(c)$ is guaranteed by the intrinsic randomness of the self-assembly process of pattern formation, thereby relying on the internal and uncontrollable manufacturing variability of the system to establish the unique input/output relation $f(\cdot)$. Noteworthy, the unique response of the PUF pattern can be probed also through other imaging techniques, such as atomic probe microscopy (AFM).

A first approach to exploit nanopatterns as physical unclonable functions is to generate a cryptographic binary code response of the system based on local features of nanoscale morphologies, as similarly reported in previous works[20,47]. In case of BCP templated patterns, the nano fingerprint feature can be converted to a binary code matrix by defining each pixel of the matrix as 1-bit or 0-bit depending on the presence or absence of minutiae in the corresponding spatial location, respectively, as reported in Fig. 3a. In other words, the binary code map can be built based on the defect map (in this case defects of the positive phase were considered). This allows the realization of the corresponding binary code matrix reported in Fig. 3b. Here, we show that the selection of the code matrix pixel size cannot be arbitrary but

should be based on morphological characteristics of the pattern. Indeed, given a pattern area, the pixel size (and thus, the number of pixels per unit area) should be selected to maximize the randomness and uniqueness of encoded binary matrices. Figure 3c reports the bit uniformity and fractional hamming distance (HD) (i.e., the percentage of bits that differs between two binary code matrices) between different binary code matrices as a function of the pixel size calculated by considering 200 different nanopattern micrographs (area of $2.38 \times 2.38\ \mu m^2$). As can be observed, the mean value of fractional HD of around 0.5 (meaning that code matrices are different and distinguishable) is achieved in correspondence of a pixel size of $238 \times 238\ nm^2$ when a bit uniformity of around 0.5 can be observed (i.e., when different binary code matrices have almost equal numbers of 0 and 1) (the relationship between fractional HD in between patterns and bit uniformity of patterns is reported in Fig. 3d). This is because bit uniformity is beneficial to achieve the maximum randomness[12], as testified by the maximum pattern entropy approaching the ideal value of 1 when bit uniformity is close to 0.5 (Fig. 3e, fractional HD between different code matrices as a function of the entropy of patterns in Supplementary Fig. S10). It is noteworthy that, a bit uniformity (and fractional HD) of 0.5 can be observed when the pixel size approximately coincides with the correlation length of BCP patterns ($\xi = 207 \pm 18\ nm$, details in Methods). Given a pattern area, it turns out that through appropriate selection of the pixel size of the binary code matrix (in this case $238 \times 238\ nm^2$), it is possible to obtain binary code matrices with bit uniformity close to the ideal value of 0.5 (in this case $0.52 \pm 0.06$) (Fig. 3f, Supplementary Fig. S11), with unit entropy close to the ideal value of 1 (Supplementary Fig. S12), and to achieve a distribution of fractional HD between different binary code matrices with mean value of $0.50 \pm 0.06$ (Fig. 3g, fractional HD for each couple of binary code matrices in Fig. 3h). The PUF randomness has been further verified through the statistical test suite NIST SP 800-22 and by analyzing the autocorrelation of bit sequences (results in Supplementary Tables 1 and 2, and Supplementary Fig. S13, respectively). All these results confirm the high randomness of the binarized code matrix obtained from local nanopattern features. In this context, the encoding capacity of the system $c^p$, defined as the number of possible responses exhibited by a random pattern where $c$ is the number of responses for each pixel while $p$ is the image area in pixels, is inherently related to the morphological properties of the nanopattern. The encoding capacity of the system can be therefore improved through two strategies: *i)* by increasing the image area (i.e., increasing the number of pixels without changing the pixel size) and/or *ii)* by reducing the pixel size of the binarized code matrices while maintaining bit uniformity and fractional inter-HD of ~0.5. Concerning the first strategy, Fig. 3i reports the maximum encoding capacity of binary code matrices ($c = 2$) as a function of the image area (by considering the selected pixel size of $238 \times 238\ nm^2$). Interestingly, an encoding capacity of the same order of magnitude of the world population can be achieved by considering an image with area below $2\ \mu m^2$, while an encoding capacity larger than the number of atoms in the known universe is achievable by considering an image area larger than $12\ \mu m^2$, showing the high density of encoding capacity of the system. Concerning the second strategy that aims to increase the encoding capacity of the system for unit area, it can be achieved by proper selection of molecular weight of involved polymers and processing conditions to increase the defect density while reducing the correlation length of the system[48] (examples in Supplementary Fig. S14). The authentication protocols based on cryptographic code obtained from fingerprint patterns PUFs can be further refined by considering not only the presence of defects in the pixel area but also defect density (Supplementary Fig. S15) and/or defect types (Fig. 3j, additional examples in Supplementary Fig. S16). In this context, it is worth remarking that the extraction of the genuine pattern through fingerprint enhancement is crucial not only to avoid bit flips in binary code matrices during the

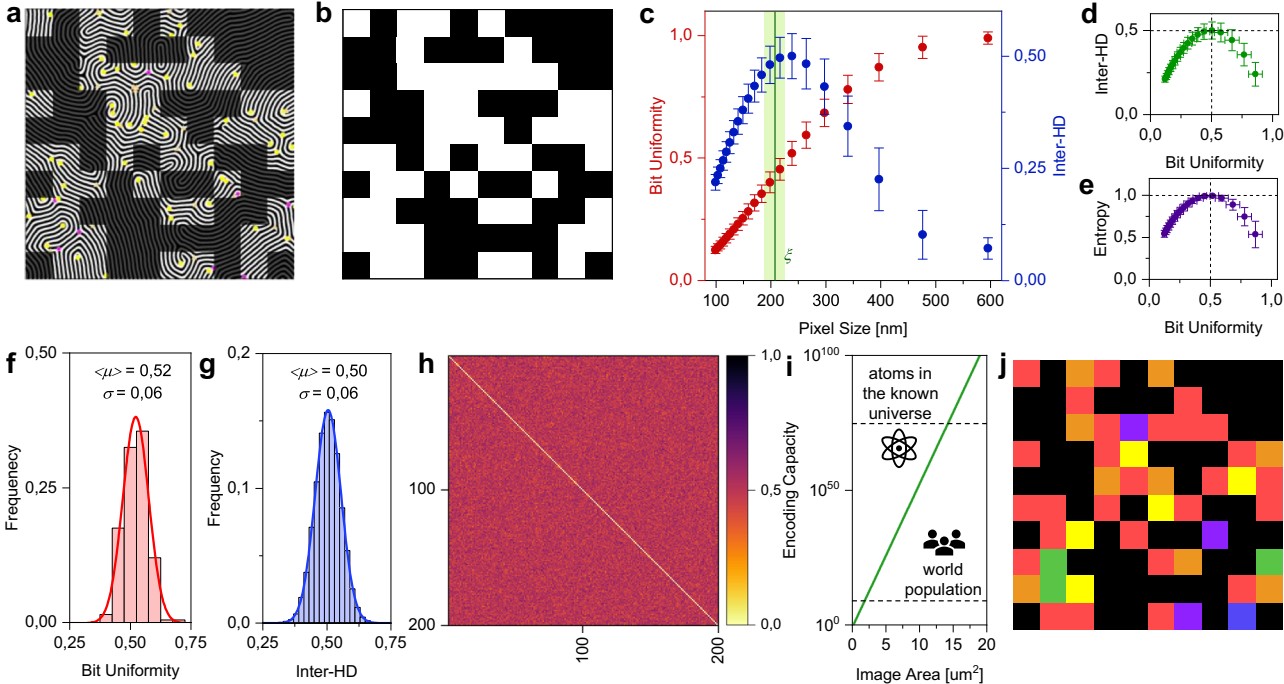

**Fig. 3 | Binary code matrices generated from nanopattern features. a** Example of binary code matrix extraction from positive phase defects of the self-assembled binarized nanopatterns (nanopattern engraved on a $SiO_2$ substrate) and (**b**) resulting binary code matrix of $10 \times 10$ pixels where white color represents the presence of a defect in the corresponding pixel area. **c** Bit uniformity of binary code matrices (red dots) and fractional HD between different binary code matrices (inter-HD, blue dots) as a function of the pixel size. Green line and shade represent the mean and standard deviation of the correlation length ($\xi$), respectively. **d** Fractional inter-HD versus bit uniformity, (**e**) unit entropy of binary code matrices versus bit uniformity (entropy was not evaluated for pixel sizes of 595 and 476 nm, since in this case it was observed a not null probability of having all 1-bit values across the matrix). In (**c**–**e**), dots are mean values while error bars represent the standard deviation calculated by considering 200 different binary code matrices obtained from different nanopattern images (area of $2.38 \times 2.38 \ \mu m^2$). **f** Distribution of bit uniformity of binary code matrices, (**g**) distribution of fractional inter-HD, (**h**) heat-map matrix representing the fractional HD between each couple of binary code matrices (HD = 0 in case of intra-HD values on the diagonal where the image is compared with itself), and **i.** maximum encoding capacity of the system as a function of the image area (code matrices with pixel size of $238 \times 238 \ nm^2$ have been considered). **j** Analog code matrix where the pixel color represents the defect type. The pixel is blue in presence of positive phase dot defects; pixel is red in presence of positive phase terminal point defects; pixel is yellow in presence of positive phase 3-way junction defects; pixel is green in presence of positive phase dot and positive phase terminal point defects; pixel is orange in presence of positive phase point and positive phase 3-way junction defects; pixel is violet in presence of positive phase dot, positive phase terminal point, and positive phase 3-way junction defects.

authentication/identification process (Supplementary Fig. S17), but also for the correct extraction of code matrices based on defect types (Supplementary Fig. S18).

## Computer vision concepts for authentication/identification

The comparison of a binarized code matrix with a database code matrix, as required for authentication/identification, implies no misalignment of the image acquired by the end user with the corresponding image stored in the database. Even small translations, rotations, and/or deformations of the end user image can cause incorrect authentication/identification due to a different conversion of the fingerprint pattern to the binary code matrix. Indeed, these transformations can cause bit flips in the related binary code matrix due to the different spatial localization of defects (Supplementary Fig. S19). Here, we show that the robustness of the authentication/ process in real-world scenarios can be enhanced by exploiting matching algorithms based on computer vision concepts, without the need of extracting a binary code matrix from the nanopattern. For this purpose, we synthesized 100 different PUF nanopatterns engraved on a $SiO_2$ substrate, and we built a database consisting of micrograph images of the corresponding fingerprint patterns acquired by a first scan (details in Methods, Supplementary Fig. S20). To obtain a set of images used for testing (test set), in a second scan we collected micrographs of the same PUF patterns. Authentication/identification were tested by comparing an image from the test set with a database

image from the database, where the genuine nanopattern is obtained through the fingerprint enhancement algorithm discussed before. Noteworthy, the obtained binarized lamellar patterns inherently endow bit uniformity close to 0.5 ($0.496 \pm 0.008$) and unit entropy close to 1 (values evaluated on the database set of images, details in Supplementary Fig. S21). For a given pair of previously binarized images, which we will refer to as database and test images, respectively, we aim at determining whether they represent the same nanopattern. As expected in real-world scenarios, slight shifts, rotations, and different magnifications are present between the database and the test set image of the same nanopattern. Given a database/test pair of images, a set of matching key points between the two images is computed through a Scale-Invariant Feature Transform (SIFT) algorithm (other methods such as Speeded Up Robust Feature, SURF, algorithm and Oriented FAST and Rotated BRIEF, ORB, were tested and gave similar results) and a FLANN (Fast Library for Approximate Nearest Neighbors)-based matcher. Then we select a few points providing the best matches (we tested between 10-15, not significantly affecting the results), and use those as a guide to build a homography transformation to map the test image on top of the database one. Examples of feature matching and overlapping of images by considering two images of the same nanopattern (example of successful matching) and on two images of different nanopatterns (example of non-successful matching) are reported in Fig. 4a and b, respectively. If the two images are taken from the same nanopattern and thus only

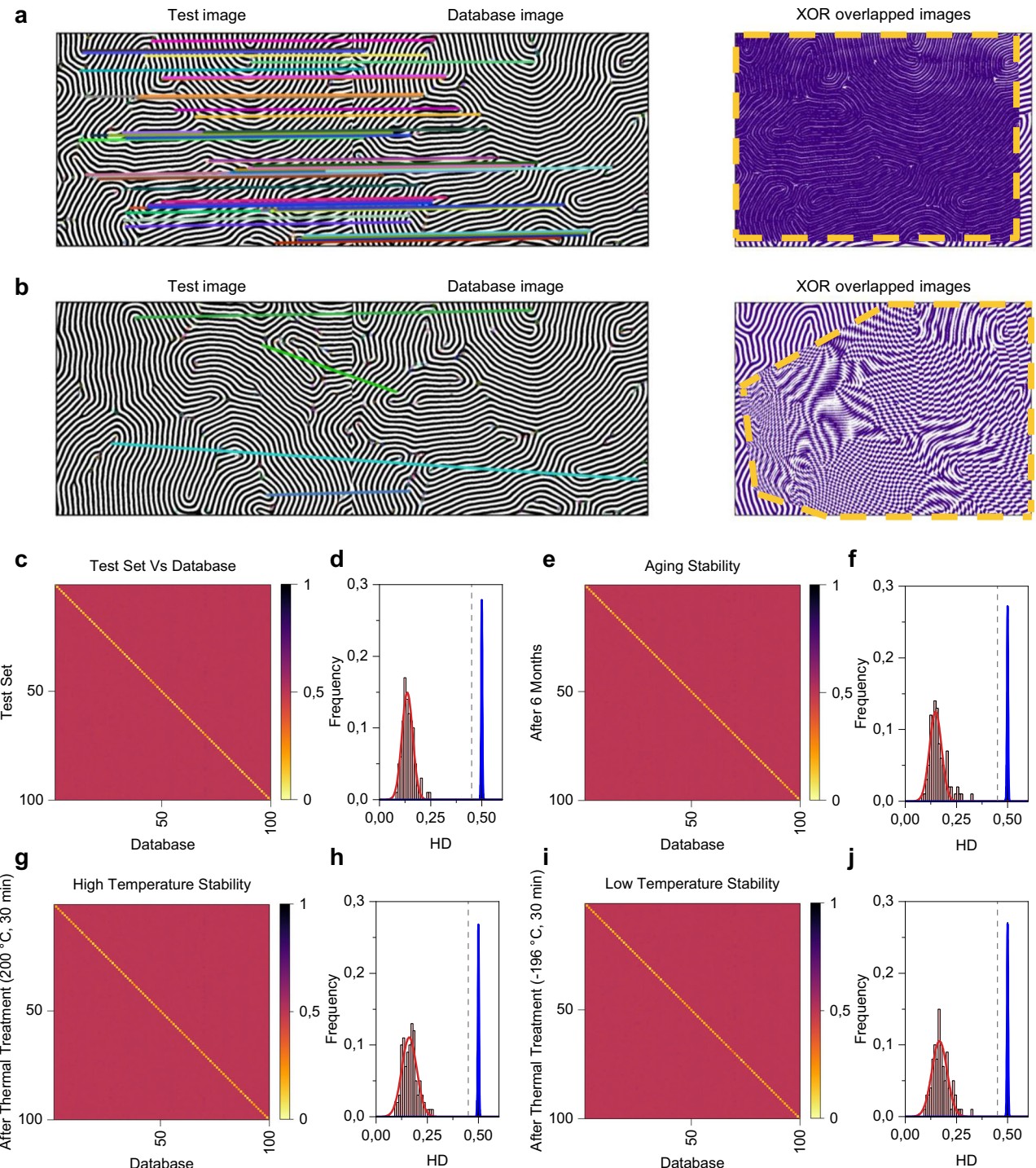

**Fig. 4 | Computer-vision-based authentication/identification of artificial fingerprint PUFs. a** Example of a successful authentication/identification and (**b**) example of a non-successful authentication/identification. These examples show the process of overlapping the test image on the database one through homographic transformation, where the overlapped image obtained through XOR operation is generated to graphically evaluate the overlapping of images under consideration. When images of the same nanopattern are compared (**a**), the XOR overlapped image is composed mainly of "0" values since the homographic transformation enables to correctly overlap the test image on the top of the database one. When images of two nanopatterns are compared (**b**), the XOR overlapped image is constituted of almost equal "0" and "1" values. Indeed, in this latter case, the homographic transformation results in a strong distortion of the test image while trying to overlap with the database one. In (**a**, **b**) the colored lines connecting the test image with the database image represent the matches of the

homographic transformation, in fact, they connect corresponding key points (represented by circles) in the two images. Dashed lines in the XOR overlapped images represent the area of effective image overlapping, i.e., the area where the test image is effectively overlapped with the database one. Heat-map matrix representing the fractional HD (**c**) and corresponding intra and inter-distributions between images from the test set and the database (**d**) images acquired after 6 months and the images from database to test aging stability (**e**, **f**) images acquired after thermal annealing of PUF devices at 200 °C for 30 min and images from the database to test high-temperature thermal stability (**g**, **h**) and images acquired after thermal annealing of PUF devices at −196 °C for 30 min and images from the database to test low-temperature thermal stability (**i**, **j**). Dashed lines in intra and inter-distributions represent the decision threshold evaluated from the comparison between the test set and the database.

differ by slight deformations (e.g., rotation, shift, etc.), the homographic transformation automatically corrects for these effects, so that the resulting image should overlap with the database one, as can be evaluated by superimposing the test image on the top of the database image through the XOR operation (Fig. 4a, Supplementary Fig. S22). On the other hand, if the two images are from unrelated samples, the transformation obtained from the matching algorithm will be unable to return an image with a reasonable overlap with the database (Fig. 4b, Supplementary Fig. S23). The quantification of the superposition is performed by considering the fractional HD calculated on the whole binary image obtained by superimposing the test image on top of the database image through XOR operation (details of the matching algorithm in Methods). Figure 4c reports the fractional HD matrix obtained by comparing images from the test set with the database. As can be observed, HD values closer to 0 can be observed when comparing images of the same PUF (i.e., the test image has a small number of bit values that differ from the database one after the homographic transformation), while HD values of approximately 0.5 can be observed when comparing images of different PUFs. A clear separation of intra and fractional inter-HD distributions can be observed in Fig. 4d, further showing the robustness of the authentication/identification protocol (mean values and std. dev. of intra and inter-HD distributions are $0.14 \pm 0.03$ and $0.500 \pm 0.003$, respectively). Here, an HD-based decision threshold of 0.45 can be identified (Supplementary Fig. S24). It is worth noticing that the extraction of the binarized fingerprint pattern from the micrograph image through the fingerprint enhancement algorithm provides noise robustness to the authentication process, a crucial aspect for exploiting image-based PUFs for real-world applications. In this context, clear separation of fractional intra and inter-HD distributions can be observed also by considering a test set of images acquired by different users operating with different equipment (details in "Methods", Supplementary Figs. S25 and S26), further corroborating the robustness of the authentication/identification protocol.

An analysis of the stability shows that PUFs can be correctly authenticated/identified even after 6 months when the sample was left in normal ambient conditions (Fig. 4e and f), revealing long-term reliable operation. Furthermore, the high thermal stability of PUFs was demonstrated by exposing samples to the harsh conditions of 200 °C for 30 min, where the PUF also experienced a high heating rate of ~15 °C s$^{-1}$ and a high cooling rate of ~2 °C s$^{-1}$ (Fig. 4g). Fractional intra and inter-HD distributions after high-temperature treatment still show clear separation (Fig. 4h). Additionally, the thermal stability of the PUF was also confirmed by low-temperature treatment at cryogenic temperature (−196 °C, liquid He) for 30 min (Fig. 4i), as revealed by fractional intra and inter-HD distributions in Fig. 4j, extending the experimental demonstration of the thermal stability of the PUF in the range from −196 °C to 200 °C. Details on intra and fractional inter-HD mean values and standard deviations evaluated for each test are reported in Supplementary Table 3. In all these cases, successful authentication/identification of all PUF patterns without any false positive or false negative has been achieved by exploiting the decision threshold of 0.45 previously evaluated through the comparison of the test set of images with the database.

## Discussion

Results show that morphological characteristics of artificial fingerprints realized with a bottom-up approach through BCP self-assembly can be exploited for the realization of physical-disorder-based PUF as anti-counterfeiting tags for authentication/identification (a comparison of the figure of merits of here reported PUF devices with literature is reported in Supplementary Table 4). While global features of the nanopattern, such as correlation length and defect density, can be deterministically controlled during the fabrication process by involved materials and processing parameters (e.g., molecular weights of the BCP, processing temperature, and time), each nanopattern is a unique entity with peculiar local features thanks to the randomness arising from thermodynamic instabilities during the self-assembly process. In this framework, results show that the coexistence of deterministic and stochastic features in the same nanopattern can be further explored to increase the encoding capacity density of the system, as revealed by the intrinsic relationship between the pixel size of the encoded binary code matrix, its bit uniformity and entropy, and the correlation length of the nanopattern.

Besides the shown possibility of using additional morphological features such as defect types to further increase the encoding density, additional morphological properties such as the line-edge roughness (LER), line-width roughness (LWR), and/or the 3D morphology and $z$-profiles obtainable through scanning probe microscopies[49], as well as morphologic parameters obtained by optical scatterometry[50], can be explored to further check the genuinity of BCP-based physical PUFs. Since it cannot be excluded a priori that a nanopattern can be physically cloned by using advanced and expensive lithographic tools with resolution <10 nm (such as, for example, through electron beam lithography and/or nanoimprint lithography, details in Supplementary Note 2), LER, LWR, and $z$-profiles of systems are unlikely to be reproducible through these lithographic techniques (details in Supplementary Fig. S27). Indeed, while the 3D structure relies on the combination of BCP self-assembly and the subsequent engraving process, LER and LWR are intrinsic properties of the BCP self-assembly process[51]. Notably, recent studies also demonstrated that LER and LWR can be engineered during the chemical synthesis of BCP, expanding the set of variables that can be explored to enhance the unclonability of PUF[52,53].

In this context, multiple morphological features can be used as a second level of authentication in a multiple-level security check scenario. This further strengthens the unclonability of BCP-based PUF that, even if the nanofabrication of similar engraved nanopatterns is theoretically possible, the cost of engineering, mass duplication, and manufacture (even for the original manufacturer that knows the exact process parameters) of another nanostructure with the same fingerprint at nanoscale accuracy makes impractical the PUF replication from a malicious party (counterfeiter)[54].

While on one hand the requirement of an electron microscope to challenge the PUF with a scanning-focused electron beam can represent an obstacle (despite recent advances in low-cost and portable SEM equipment), on the other hand, the requirement of specialized equipment can further enhance the security of the authentication/identification process. Moreover, these nanometer-scale artificial fingerprints can be exploited for anti-counterfeiting applications by taking advantage also that these tags are difficult to localize without an apriori knowledge of their location.

Notably, the acquisition of these image-based PUFs through the supply chain can be performed by means of numerous affordable tabletop SEM instruments available in the market, capable of measuring very large and heavy objects with sufficient resolution to detect and extract the fingerprint pattern through the fingerprint enhancement algorithm[55]. In any case, the acquisition of the fingerprint nanopattern through SEM imaging requires the insertion of the object inside a space-limited vacuum chamber. As an alternative, instruments such as AFM can be employed to acquire the nanopattern with sufficient image quality and resolution in air, without the need for the insertion of the object inside a vacuum chamber. In this framework, it is worth noticing that AFM and SEM provide complementary insights on the nanopattern, as each technique addresses the same measurand through different physical principles (i.e., tip-substrate interaction and electron beam-sample interaction, respectively) as discussed in ref. 56. This complementarity can be further exploited to check the authenticity of the physical nanopattern.

Differently from previous works where nanoscale tags are affixed to the substrate of interest[8,12,15,18–20,57], the here proposed artificial fingerprint is directly engraved on the target substrate and, thus, cannot be removed (unless the substrate is mechanically scratched, but this will leave marks that can be considered as a trace of counterfeiting). As an important consequence, the PUF robustness and environmental stability are inherited by the intrinsic physical/chemical/mechanical properties of the material in which the fingerprint pattern is engraved. Thus, the proper choice of the target material is crucial to ensure stability of the PUF and to prevent the impairment of the authentication/identification processes. This means that tailored choice of substrate would enable the realization of PUF devices working even when exposed to harsh conditions (high/low temperatures, exposure to chemicals, mechanical agents, radiations, etc.) as required, for example, for space applications. In this context, it is worth mentioning the stability of the self-assembled BCPs upon substrate stretching[58], mechanical deformation, or crumpling[59].

Depending on the specific target application, a viable alternative to the engraving process is represented by sequential infiltration synthesis (SIS) that directly enables the transformation of the soft polymeric fingerprint nanopattern into an inorganic nanopattern characterized by high thermal stability[60] (examples of $Al_2O_3$ and $TiO_2$ nanopatterns realized through SIS are reported in Supplementary Fig. S28). Even if the same fingerprint enhancement algorithm for pattern extraction and the same authentication/identification protocol can be used, note that in this case the pattern is not engraved on the target object but is superimposed on the surface of the object of interest.

Furthermore, it is worth noticing that BCP-templating is compatible with CMOS technology[61], allowing the integration of these PUF devices with conventional electronics. As an example, the possibility of realizing fingerprints directly on chips (or on the chip packaging) to assess their legitimacy can be crucial to face back doors and hidden functionalities of fraudulent clone chips that, in the framework of global shortage of chips, can threaten global security[62]. The high versatility and advanced maturity of the BCP lithography allow extending the described process to several materials, including metals, ceramics, and polymers[37], enabling the use of these identifiers for a vast range of applications. In perspective, these image-based PUF that rely on artificial fingerprints can be included in conventional tags (such as QR codes) in the context of a multiple-level security strategy, as well as can be integrated with optical and/or electrical readout to generate a larger amount of unpredictable challenge/response pair as required for remote authentication applications and secret key generation (similarly to the approach reported in ref. 46). All these characteristics enable artificial fingerprints based on BCP-templated substrates to comply with security property basic requirements in terms of robustness and physical unclonability for commercial applications, taking advantage of the intrinsic low-cost of BCP templating on a large scale that easily allows mass production.

In conclusion, we show that artificial fingerprints as anti-counterfeit identifiers can be engraved on a wide range of target substrates including—but not limited to—diamond, Si, SiO$_2$, and quartz to realize disorder-based PUFs readable through electron microscopy. By proposing a user-independent strategy for successfully extracting the genuine nanopattern even from low-quality micrographs that provide robustness to the authentication process, we show that engraved nanopatterns can be encoded in binary code matrices representing a unique bit stream of information that endow high bit uniformity, high uniqueness, and high entropy. In this context, we show the importance of an appropriate encoding of nanopattern in the corresponding binary code matrix depending on morphological nanopattern features, including defects and correlation length of the self-assembled nanostructures that significantly affect the security properties of the PUF. In addition, we propose a computer vision-based strategy that enables robust authentication/identification processes in a real-world scenario, as demonstrated by testing 100 PUF devices. Through this strategy, long-term reliable operation and robust authentication/identification against thermal treatment (range −196 °C to 200 °C) have been demonstrated. These results pave the way for the realization of PUF that embraces the inherent stochasticity of nanoscale self-assembled materials as a randomness source.

## Methods

### Materials

The synthesis of $\alpha$-hydroxy $\omega$-Br polystyrene-*stat*-polymethyl methacrylate (PS-*stat*-PMMA) random copolymer (RCP) with $M_w = 14.60$ kg mol$^{-1}$, styrene fraction ($f_{PS}$) of 0.59 and polydispersity index (PDI) 1.30, is described elsewhere[48]. BCP with molecular weight of $M_w = 42$ kg mol$^{-1}$, $f_{PS} = 0.50$, and PDI = 1.07 (purchased from Polymer Source Inc. and used without further purification) was exploited for the realization of nanopatterns reported in Fig. 1d and e as an example of typical defects in lamellar BCPs, associated with a pitch of ~ 27 nm. The patterns in Fig. 1f–i, Supplementary Figs. S1-S9, S14, S18, S19, S22, S23, S27, S28, and the PUF devices (Figs. 2,3,4, and Supplementary Figs. S20 and S25) were fabricated by considering BCP with $M_w = 66$ kg mol$^{-1}$, $f_{PS} = 0.50$ and PDI = 1.09 (purchased from Polymer Source Inc. and used without further purification), that results in nanostructures with a pitch of 38.2 ± 0.3 nm (this value represents the weighted mean of the pitch value and its error calculated by analyzing the first peaks of the fast Fourier transform over 200 images). Note that this value is consistent with one previously reported literature for BCP with the same molecular weight[48,63]. Toluene (99.8% anhydrous), acetone (99.5% anhydrous), and isopropyl alcohol (98% anhydrous) were purchased from Sigma Aldrich. Polymethyl methacrylate (PMMA) A4 positive electron-beam resist was purchased from MicroChem. Methyl isobutyl ketone (MIBK) developer was purchased from KemLab.

### Fabrication of fingerprint patterns by block copolymers

The surface of different materials (silicon, silicon oxide, quartz, diamond, Cu foil, and nickel-plated brass) were functionalized by RCP grafting process to promote the perpendicular orientation of lamellar BCP nanostructures. First, the substrates were cleaned in an ultrasonic bath in acetone followed by isopropyl alcohol and functionalized by O$_2$ plasma treatment at 130 W for 20 min. Then, a solution of RCP (18 mg in 2 ml of toluene) was spin-coated for 60 s at 3000 rpm onto the functionalized substrates. The grafting process was performed in a rapid thermal processing (RTP) machine Jipelec JetFirst200 at high temperature ($T_a = 290$ °C) for an annealing time ($t_a$) of 300 s, in an N$_2$ environment with a heating rate of 15 °C s$^{-1}$. Automatic cooling to room temperature was set to 240 s. The non-grafted polymeric chains were then removed by sonication in toluene for 6 min, resulting in a final grafted RCP layer thickness of ~7 nm, as measured by spectroscopic ellipsometry (alpha-SE ellipsometer from J.A. Wollam Co.). BCP solution (18 mg in 2 ml of toluene) was then spin-coated over the RCP functionalized surface at 3000 rpm for 60 s resulting in a total BCP thickness of 35 nm. The self-assembly was promoted by RTP at 230 °C for 600 s in a N$_2$ environment with a heating ramp of 15 °C s$^{-1}$ and automatic cooling to room temperature for 240 s. Selective removal of the PMMA phase of self-assembled BCP was achieved by exposure of the samples to ultraviolet radiation (5 mW cm$^{-2}$, $\lambda = 253.7$ nm) for 180 s followed by isotropic O$_2$ plasma etching (40 W for 30 s).

### Engraving nano fingerprints on target substrates and materials

The pattern transfer of the fingerprint pattern onto the substrate surface was performed by reactive ion etching (RIE) in a PlasmaPro 100 Cobra 300 ICP (Oxford Instruments Plasma Technology) with different chemistries depending on the etched material. Silicon etching was achieved by SF$_6$/C$_4$F$_8$ RIE with ICP plasma of 750 W and RF table of

35 W. Silicon oxide and quartz etching was achieved by $CHF_3$/Ar RIE and power of 35 W. Diamond etching was achieved by $O_2$ RIE with ICP plasma of 2000 W and RF table of 200 W.

## Morphological characterization of nanopatterns

The micrographs of artificial fingerprints were acquired by FEI Inspect-F field emission gun scanning electron microscope (FEG-SEM) using an Everhart-Thornley secondary electron detector (ETD). Additionally, a dual-beam SEM FEI Quanta 3D FEG was employed for authentication/identification of artificial fingerprints, enabling the collection of the SEM micrographs through alternative instrumentation.

## Fingerprint enhancement algorithm for nanopattern binarization

The fingerprint enhancement algorithm exploited for pattern binarization was adapted from the algorithm described in ref. 44 by exploiting the `fingerprint_enhancement` Python library. Parameters of the `enhance_Fingerprint` function were adapted on the basis of the pitch of the lamellar structures, obtained from the Fourier transform of the corresponding nanopattern image, and the resolution of the SEM image. The fingerprint enhancement binarization was compared to standard binarization techniques implemented through auto-local thresholding methods with default parameters of ADAblock[45].

## Characterization of artificial nano fingerprint properties

Spatial coordinates, types, and density of topological defects of binarized nanopatterns, as well as the analysis of correlation length, was performed through the automated defect analysis tool ADAblock described in ref. 45, modified to be integrated into an automated analysis process including the binarization through fingerprint enhancement algorithm. The analysis of Fig. 3 was performed over 200 nanopattern SEM images with an original area of $2.64 \times 3.08\,\mu m^2$ acquired over different areas of the sample, while the correlation length of nanopatterns was evaluated by considering 11 images with an area of $4.14 \times 3.56\,\mu m^2$ of the same nanopatterned sample (here, only images with a reduced magnification were used to have a higher grain size/image dimension ratio as required for the analysis of correlation length). The binary code matrices have been obtained from 200 SEM images of nanopattern micrographs by considering a central square area of the SEM image of $2.38 \times 2.38\,\mu m^2$. The analysis of Fig. 2 was performed over about 30 images for each binarization technique, in fact, a couple of these images were discarded. The defect density reported in Fig. 2e is expressed as the median value of the density for the different defect types evaluated over about 30 images.

The bit uniformity ($p_i$) of the $i$-th binarized code matrix, defined as the distribution of ones and zeroes in the bit stream, has been calculated through the equation:

$$p_i = \frac{1}{n}\sum_{k=1}^{n} R_{i,k} \qquad (1)$$

where $R_{i,k}$ is the binary bit response of the $k$-th pixel of the $i$-th binarized code matrix, while $n$ is the number of pixels of the code matrix. A value of unity means that all responses $R_{i,k}$ of the $i$-th pattern are ones. The unit entropy of the $i$-th binarized code matrix has been evaluated through the equation[19]:

$$E_i = -\left[p_i \log_2 p_i + (1-p_i)\log_2 p_i(1-p_i)\right] \qquad (2)$$

## Fabrication of PUF devices based on artificial fingerprints

Nanoscale PUF devices with dimensions of $3\,\mu m \times 3\,\mu m$ were realized by creating a metallic frame surrounding the area of interest of the previously nanopatterned substrate, as shown in Supplementary

Fig. S20. For this purpose, the area of interest was defined by electron-beam lithography (EBL) in a dual-beam SEM FEI Quanta 3D FEG on PMMA A4 positive-tone resist on the patterned $SiO_2$ substrate. After the development with MIBK:IPA (3:1) developer solution, a double layer of Ti/Au (2 nm and 3 nm of thickness, respectively) was deposited by RF sputtering followed by a lift-off process under sonication. A discussion on alternative methods for the fabrication of the "find-me structures" to define the area of interest can be found in Supplementary Note 3.

## Computer vision-based matching algorithm

The matching algorithm has been implemented by exploiting computer vision algorithms of OpenCV python library. As the first step, the image was cropped of $1.04\,\mu m$ on each side of the SEM image to remove the Au frame of the PUF device that can interfere with the binarization process. The resulting image has an area of $1.47 \times 2.06\,\mu m^2$ or $1.49 \times 2.07\,\mu m^2$ on which to perform the following analysis. Then, the homographic transformation, required for overlapping the test image on top of the database image, was calculated through SIFT (SURF or ORB were tested to give similar results) and a FLANN-based matcher from OpenCV python library. Note that, after performing the homographic transformation, the transformed image (no longer binary) is binarized to 1 and 0 values through a thresholding technique, where the threshold was chosen halfway between the minimum and the maximum values. Then, quantification of the superposition was performed by calculating the fractional HD between considered images (calculation was performed over $987 \times 708$ or $996 \times 717$ pixels). Overlapped images in Fig. 4a and b have been obtained by superimposing a test image after homographic transformation and the database image with XOR operation between the two. Note that XOR operation returns a 0-pixel value if the corresponding pixel of the test and database image is characterized by the same bit value, 1-pixel value otherwise. In these terms, an overlapped image with a high number of 0 values represents a good overlapping between test and database images. Note that performing fractional HD is nothing but performing the XOR operation over two images, then summing all pixel values of the obtained overlapping image and normalizing over the number of pixels. Given the sharp nature of the binarized images, we typically obtain very clean cancellations when a good match is found, consistently leading to a clear separation of the fractional HD between good and bad matches.

## Tests of PUF robustness

The high-temperature thermal stability of PUF devices was tested by subjecting the samples to heat treatment by rapid thermal annealing at 200 °C for 30 min. The annealing was performed in a RTP Jipelec Jet-First200 in $N_2$ environment with and heating rate of $15\,°C\,s^{-1}$ which puts the device into even harsher conditions when compared to conventional furnace heating. Then, an automatic cooling to room temperature was set to 240 s. The low-temperature thermal stability was tested by subjecting the sample to a liquid He bath at −196 °C for 30 min with a cooling rate of - $2\,°C\,s^{-1}$. Aging stability was evaluated by collection of the SEM images of the same set of devices after 6 months from its fabrication, by storing devices at normal ambient conditions.

## Data availability

The data that support the findings of this study are available on Zenodo (https://doi.org/10.5281/zenodo.13384908). All other data are available from the authors.

## Code availability

The codes are available on Zenodo (https://doi.org/10.5281/zenodo.14179662) and can also be accessed on GitHub (https://github.com/ChiaraMagosso/NanoIdentifiers).

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

## Acknowledgements

Part of this work was supported by the European project MEMQuD, code 20FUN06. This project (EMPIR 20FUN06 MEMQuD) has received funding from the EMPIR program co-financed by the Participating States and from the European Union's Horizon 2020 research and innovation program. Part of this work was supported by the European project OpMetBat, code 21GRD01. The project has received funding from the European Partnership on Metrology, co-financed from the European Union's Horizon Europe Research and Innovation Programme, and by ParticipatingStates. Part of this work has been carried out at Nano facility Piemonte INRiM, a laboratory supported by the "Compagnia di San Paolo" Foundation, and at the QR Laboratories, INRiM. Part of this work was supported by the European Union—-NextGenerationEU under the National Recovery and Resilience Plan (NRRP), Mission 04 Component 2 Investment 3.1|Project Code: IR0000027–CUP: B33C22000710006–iENTRANCE@ENL: Infrastructure for Energy TRAnsition aNd Circular Economy @EuroNanoLab. The authors acknowledge Dr. Bruno Torre (INRiM) for his kind support on AFM measurements, and "Gioielleria Mariatti Torino" for providing some substrates tested in this work.

## Author contributions

I.M., C.M., F.F.L., and G.M. conceptualized the idea and designed experiments. I.M. performed sample preparation and experimental characterization. C.M. developed codes for image analysis, data analysis and characterization of PUF performances. I.M. and C.M. supported by F.F.L. and G.M. performed data analysis. M.F. supported experimental activities. S.C. supported code development. F.F.L and G.M. provided funding and supervised the research. I.M., C.M., F.F.L., and G.M. wrote the manuscript. These authors jointly supervised this work: F.F.L, G.M. All authors participated in the discussion of results and revision of the manuscript.

## Competing interests

The device configuration and implementation method of artificial fingerprints are currently under patent filing (PCT/IB2024/059078). Patent applicants: Politecnico di Torino, Istituto Nazionale di Ricerca Metrologica. Inventors: I.M., C.M., F.F.L., and G.M. I.M., C.M., F.F.L., and G.M. declare no other competing interests. S.C. and M.F. declare no competing interests.
