## [Transparent Peer Review file · Nature Communications]

Artificial fingerprints engraved through block-copolymers as nanoscale physical unclonable functions for authentication and identification

Corresponding Author: Dr Gianluca Milano

Version 0:

Reviewer comments:

Reviewer #1

(Remarks to the Author)

Title: Artificial fingerprints engraved through block-copolymers as nanoscale physical unclonable functions for authentication and identification

In this paper, the authors reported disorder-based PUFs for anti-counterfeiting using nanopatterns formed by block copolymer self-assembly. The authors propose a user-independent strategy for successfully extracting nanopatterns from low-quality micrographs and show that nanopatterns can be encoded into binary code matrices that provide high bit uniformity and uniqueness. In addition, the authors also discussed the appropriate encoding of morphological nanopattern features, including defects and correlation lengths. Finally, the authors propose a computer vision-based strategy that enables robust authentication processes. Unfortunately, however, this study seems challenging in obtaining novelty in terms of block copolymer nanopattern-based PUF label formation and authentication using minutia, and significant improvement in achieving novelty is necessary. I expect this manuscript to become an outstanding paper after revision.

(1) Several studies have reported using self-assembly of polymers to produce authentication tags. In this respect, the pattern manufacturing method and authentication elements shown in this study are similar to previous results. Therefore, if the paper's novelty is to increase accuracy through image processing, additional explanation is needed, especially in the abstract.

(2) Since the fingerprint tag produced in this study is characterized by nano size, expensive and space-limited equipment such as SEM is required to identify the morphology for authentication; therefore, to discuss actual application to the supply chain as shown in figure 1b, it will also be necessary to discuss difficulties that may arise in measurement or actual use.

(3) The explanation in Figure 1f-i states that the self-assembly engraving technology can be applied directly to the surfaces of various products that are not flat, such as diamonds and cameras. However, it is technically very difficult to form the BCP self-assembled structure on an uneven surface and etch it uniformly. Therefore, explaining the technologies and problems required to apply engraving technology is necessary. Furthermore, regarding the SEM image inserted in Figure 1f-i, the figure may need to be modified as it may cause readers to misunderstand that the technology was applied on various product surfaces that are not flat.

(4) In line 204th, the authors explain that the intrinsic randomness of the self-assembled structure can guarantee the uniqueness of the response of the PUF. However, the uniqueness of the response is expected to be affected by several factors, such as the deviation of the response depending on the characteristics or type of challenge input to the PUF and the averaging effect of the response depending on the size of the PUF label. Therefore, it seems necessary to further explain that the self-assembly's intrinsic randomness guarantees the response's randomness.

(5) In Figure 2, the authors achieved improved image recognition capabilities through fingerprint enhancement, allowing different types of defects to be distinguished much better. However, in Figure 3, the key is generated only through the

presence or absence of minutia and density. Therefore, discussing the strategies to actively utilize the advantages of improving image recognition ability through data processing in key generation, such as encryption of specific minutia types, will strengthen the authors' argument.

(6) The thermal stability discussed in Figure 4e will vary greatly depending on the constituent materials of the product to be engraved or heat treatment conditions. For example, if a fingerprint is engraved on a product made of ceramic, it will be thermally stable even in a high-temperature environment of 300 degrees, but in the case of a polymer, it is questionable whether it will be thermally stable under the same conditions. In this regard, explaining the importance of material selection to ensure thermal stability and the temperature limit that can maintain thermal stability would be better.

(7) Strictly speaking, the concept of PUF is an object that performs authentication with a pair of inputs called 'challenge' and an output called 'response' without exposing the internal structure. In this respect, the fingerprint pattern fabricated by the authors is close to an anti-counterfeit tag or identifier because authentication must be performed by verifying the internal structure. Therefore, if the authors briefly mention that the fingerprint pattern is close to the anti-counterfeit identifier, the author will be able to prevent confusion among readers.

(8) It would be better to write the full name before using the abbreviated form in the text. For example, 'RCP' in the method part is supposed to mean random copolymer, but it may need to be clarified for those unfamiliar with the content. Reaffirming the format and typos of the paper will help readers understand. For example, in line 150th, there is no period, and in line 422, 'counterfeiting' should be 'anti-counterfeiting.'

Reviewer #2

(Remarks to the Author)

Murataj et al describe a study of using fingerprint patterns from block copolymers as nanoscale physically unclonable functions (PUFs) for authentication. The idea is very interesting. Block copolymers provide a pretty large range of random structures. The manuscript does a very thorough job of analyzing images of the fingerprint patterns and developing a robust method for a matching algorithm and analysis of the entropy of the structures/defects in block copolymer patterns. They also transferred patterns to a variety of substrates and evaluated the physical robustness of the patterns over time and temperature. There are two main limitations to the work. The verification process is very difficult (requires a high-resolution scanning electron microscope (SEM) with the nanopattern accessible) and they do not discuss if the nanopatterns are in-fact unclonable (I do not think they are). The block copolymer structure as described is more like a high entropy nanoscale QR code than a PUF. I discuss these two topics in more detail below.

Verification/reading of the PUF function requires a scanning electron microscope to image the nanopattern. The manuscript describes the difficulty of authenticating the PUF as an enhancement to the security. I consider it to be a severe limitation to the use of the PUF. It would work much better and be more usable if there was an easier way to authenticate the PUF function. Putting your chip into an SEM is pretty involved, especially after the chip has been packaged. In that case you would have to open up the package and destroy the chip. As described, it is only really readable during wafer fabrication. In general there is pretty good security during the wafer fabrication process and the main counterfeiting concern is during packaging and post-packaging. It would be much better if there was an electrical or optical method for measuring the nanopattern that would get out a unique signature from the PUF. An electrical or optical method might also be more sensitive to nanoscale details of the nanopattern than the SEM imaging and defect analysis is. It also could be sensitive to the materials and the nanoscale 3D structure of the nanopattern. These additional factors would make it much more difficult to clone the PUF.

The proposed structures are not actually unclonable as described. If the nanopattern is available for SEM imaging, it could be cloned. The manuscript only addresses this concern by saying that the PUF could be hidden on the chip. This isn't really a good security feature because the location is difficult to protect and could be discoverable if the PUF is widely used. There are two methods I'm aware of that could clone the proposed PUFs. The first method is nanoimprint lithography (NIL). NIL can be used to make a replica of the PUF nanopattern and transfer it to a new chip (or many new chips). Their verification/reading method using imaging SEM is not sensitive to the depth of the pattern or the material that the pattern is made out of. They are simply analyzing the 2D binarized fingerprint structure and defects on the 20-50 nm length scales. These defects could easily be replicated with the nm-scale resolution of NIL. The second method is e-beam lithography. Since the PUF nanostructure is imageable, it would be possible to use the digital image to create a pattern for e-beam lithography. The 2D pattern with sub-20 nm resolution would require a very good e-beam tool and some process development, but would be doable if the authentication is simply through comparison of large defects in the fingerprint pattern. To make the nanopattern unclonable, it needs to be buried within the chip structure to protect it from imaging, and a non-imaging method of verification/reading would need to be developed. If an electrical/optical method was sensitive to nanoscale fluctuations in the 3D nanostructure it would be effectively unclonable since these features are much more difficult to fabricate controllably even if the PUF was imageable.

Reviewer #3

(Remarks to the Author)

This paper presents a novel physically unclonable function (PUF) manufacturing technique that exploits block-copolymers (BCPs) to form unique nanopatterns and precisely copy these patterns onto multiple substrate materials through pattern

transfer techniques. This process not only takes advantage of the inherent randomness of nanoscale material self-assembly to generate complex nanostructures that are difficult to replicate or clone, but also, through the developed algorithm, encodes these nanopatterns into binary code matrices with high entropy and high uniqueness, which greatly increases the coding density and enables large amounts of information to be stored in a very small area. In addition, the authors propose a computer vision-based authentication and identification strategy to further enhance the PUF's security and feasibility. The experimental results validate the high reliability of the fabricated PUF, under aging and high temperature conditions. Features extracted from scanning electron microscope (SEM) images through computer vision technology were successfully converted to binary codes that are more suitable for security authentication. Several detailed comments are provided as follows:

1. While the paper discusses the inherent stochasticity of self-assembled block copolymers, it would be necessary to include a more comprehensive analysis of the randomness of the nanopatterns and PUF. It is expected to add more test results to further verify the PUF's randomness, such as autocorrelation function (ACF) test and NIST SP 800-22 test results.
2. How is the reliability performance for the fabricated PUF under the low temperature, such as the range of -40°C ~ 0°C ?
3. There is no direct comparison with other existing materials or techniques applied for PUF, please make some comparison with the state-of-the-art PUF designs in terms of the widely-accepted figures of merit, such as reliability, randomness, uniqueness, normalized bitcell area, etc.

Version 1:

Reviewer comments:

Reviewer #1

(Remarks to the Author)

I appreciate the quick work the authors did in addressing my concerns with the first manuscript and the effort the authors made to address each of my concerns in their response to my initial review. The work presented here is truly impressive. In conclusion, I have no further suggestions for improvement.

Reviewer #2

(Remarks to the Author)

The authors have revised the manuscript and added additional data and discussion. They have added additional more detailed information on the entropy of the structure and measurement repeatability and completed a detailed statistical analysis based on a NIST standard. This work has improved the manuscript a lot. The fundamental remaining problem with the manuscript is the combined difficulty of fabrication and imaging/verification of the nanostructure. Putting an object in an SEM is not an easy way to verify the PUF. It would be much better if they had an easier method like optical scatterometry (OCD).

The response says that the nanopattern can be left exposed to be accessible for imaging. Doing that for a chip is quite involved. For a chip, it means that the package would have to be partially removed to open access to the pattern. That would break the hermetic seal of the packaging and risk environmental exposure to the chip. It also requires the pattern to be made on the chip at the back end of line on top of the interconnect wiring (or a window needs to be etched into the interconnect layers to reveal a deeper layer pattern). Additionally the nanopattern requires an additional lithography step to pattern the nanopattern to the 2-4 μm square. The target has to be small and well-defined or it will be really difficult to ensure that the same spot is measured as was originally certified. The manuscript suggests that the engraved nanopattern could be put on the outside of the final product. Opportunities for that are somewhat limited since the surface where you put the nanopattern would have to be quite smooth. They discuss putting the pattern on a curved surface. That surface on a 1-2 μm length scale is relatively flat and if it was glass it would also be very smooth. Note though that if the roughness is more than about 5-10 nm it will obscure the nanopattern. Typical metal and plastic surfaces are much rougher than that and would be unsuitable for a nanopattern PUF. For microelectronics the packaging is done at a different location than the wafer fabrication, so a PUF on the packaging is not as useful.

The manuscript and supplemental do not appear to ever state the pitch of the block copolymer patterns and the microscope images are not consistent. Figure 1 appears to make the pitch to be about 80-100 nm (caption states that the scale bar is 400 nm and has 4 lamella in the right side and 6-7 in the left side). Figure 2 appears to make the pitch to be 33 nm (caption states the scale bar is 200 nm and has 6 lamella). Figure S2 has 7 lamella with scale bar of 200 nm (pitch of 29 nm). The AFM image in figure S24 for the cross section shows 15 lamella in 600 nm (pitch of 40 nm). They should report the pitch and check their scale bars. The methods section only lists one BCP, so one assumes all the pitches should be similar. If multiple BCPs are used, they should report that.

Their analysis method is based on defects and fingerprint analysis. This is looking at features on the 10+ nm scale (assuming a BCP period of 30 nm). They state that one could include the nm-scale roughness of the lines in the algorithm to increase the security and entropy of the pattern. I would be very concerned with SEM repeatability if you included the line roughness in the algorithm. There is considerably work in the semiconductor industry on very expensive fab CD-SEMS to get consistent line roughness values. It is strongly dependent on the imaging conditions and how you process the data and focused on getting statistical information on roughness across a large image with lots of lines. With that approach you lose the unique information from the spatial distribution of roughness that would be required to use it towards the PUF. Repeatedly measuring the spatial fluctuations and distribution of roughness across PUF pattern would be very difficult.

The manuscript states that AFM could be used for looking at line roughness and the vertical dimensions of the pattern. AFM is pretty poor at nm-scale edge roughness due to tip shape effects. It would be very strongly dependent on the current tip shape and prone to artifacts. The pattern would also have to be relatively shallow for the tip to be able to fit into the ~15 nm wide trenches between the lines and reach the bottom. Most of the line shape in figure s24 is due to tip artifacts and not due to the actual shape of the engraved lines. It wouldn't be very useful for validation of a nanoscale PUF.

Nanoimprint is pretty good at replicating structures. If someone had access to the PUF pattern, you could create a replica and master stamp with ~nm-scale matching of the PUF including the roughness. The main limitation to nanoimprint resolution is the original lithography of the master template. Nanoimprint tools are also relatively inexpensive.

Reviewer #3

(Remarks to the Author)

All of my previous comments have been addressed.

Version 2:

Reviewer comments:

Reviewer #2

(Remarks to the Author)

I thank the authors for their additional responses and for correcting the errors in their scale bars in the manuscript. Several of their responses are in error. These are discussed below:

The authors are in error in their comments on NIL. I'm not sure where the reported resolution of 50-100 nm for NIL came from or what is meant by resolution (discussed more below). NIL is very good at replicating structures. The biggest challenge for NIL in nanofabrication is fabrication of the original high-resolution master. In the case of their PUF, that step is already completed and just needs to be replicated. A recent review article by SV Sreenivasan (doi:10.1038/micronano.2017.75), one of the pioneers of NIL and a founder of Molecular Imprints (now Canon), says that the resolution of NIL is <3 nm (referencing work templating from carbon nanotubes). He also cites studies where sub-7 nm half pitch structures have been fabricated with NIL. He shows commercial examples for hard disks where companies routinely obtain sub-10 nm patterning rapidly in their production fabs. Note that the definition of resolution here is describing the half-pitch of the lines being printed, not the line edge roughness or sub-features of the lines that can be replicated. These 7-10 nm half pitch lines are less than half the width of the lines fabricated from the BCP PUF in this manuscript. Cloning with NIL will not be that difficult. Also, the resolution from a cheaper NIL system isn't going to be that worse than an expensive one. The big advantage of expensive systems is the ability to go faster with fewer defects and print much larger areas through methods like step and flash. The PUF is very small, so most of the complications of NIL will not apply. The authors also discuss challenges with NIL overlay. Overlay is unnecessary to clone the PUF. The frame around the PUF would be part of the imprint pattern and not need overlay. The alignment of the PUF with the object does not matter for authentication unless they incorporate a fiducial fabricated during a different process step. The example of sub-10 nm NIL lines for hard drives was actually one of the first use cases for BCP DSA. E-beam lithography wasn't good enough so HGST combined density multiplication DSA with an e-beam pattern to get smaller pitch structures than what they could directly print with e-beam to make their NIL masters for bit-patterned media. Now they use SADP with ebeam. Defectivity challenges for NIL are for printing large area patterns. One can be pretty defective and still get good yield on 2 um areas. It would also be easy for the malicious actor to make a bunch of PUFed counterfeit items and then use an SEM to reject the ones with defects.

The authors are also in error on their comments on CD-AFM and the ability to measure 3D morphology. CD-AFM is done on isolated lines or on lines with large space between them (<https://doi.org/10.1117/1.JMM.15.4.044006>). The patterns in this manuscript are 38 nm pitch with nominal trench width of 19 nm. CD-AFMs for nanoscale lines use a tip with a flare on the end to prevent the tip sidewall from touching the line. This makes them wider than super sharp AFM tips (often 20-50 nm). The tip also has a horizontal oscillation to do side wall profiling. The width of the tip also smooths the measured high frequency LER. Very narrow trenches require using a very high aspect ratio probe. Work has been done using carbon nanotube tips or other HAR tips. Any of the HAR probe options introduce tip flexure and likely hit the sidewall somewhere other than the end of the tip. They are very challenging to use and get repeatable results side wall profiling with CD-AFM. Conventional AFM with a 5 nm tip diameter will show you the top structure of the 38 nm pitch fingerprint patterns convoluted with tip shape. It will struggle to reach the bottom if the depth is more than the trench width. In that case it will only touch the bottom in the middle of the trench. The sidewall shape will be dominated by the tip. The 5 nm diameter is only at the end of the tip. Beyond that is a pyramidal shape that quickly tapers at an angle shallower than the sidewall angle of the line structures. 5 nm diameter tips will also smooth the high frequency LER.

Their discussion on surface roughness doesn't completely make sense. What matters is the nanoscale roughness within the 2 um area of the PUF. A curved surface could still be smooth locally on the nanoscale. It doesn't surprise me that a polished optical lens would have lots of areas locally smooth enough for BCP self assembly. What was the local surface roughness in the 4 um region in figure R2? The 104.3 nm RMS value for the optical profilometer was over a 35x42 um field with a method

with micron resolution. The surface slopes are pretty smooth so a lot of local areas might not have a lot of nanoscale roughness. One conceivably could identify a smooth enough spot for the PUF or use some sort of local polishing method to make a smooth spot. They should try something like a rolled metal foil. Nanoscale roughness will affect the ability of the BCP to form vertical lamella. You likely will end up with mixed orientation of the BCP. Thicker BCPs will probably make it worse because you would get multiple orientations in the depth of the film and the pattern transfer would be poorly defined. High frequency roughness will affect the ability to transfer the pattern to the substrate. It is possible as the authors point out that mixing of the high frequency substrate roughness with the BCP fingerprint will add to the entropy in some cases. It also will make it more difficult to cleanly image the resulting fingerprint. The referenced paper for tall lamella is using a multistage directed self assembly process to get the tall lamella. That process wouldn't work on really rough substrates.

(Remarks on code availability)

Version 3:

Reviewer comments:

Reviewer #2

(Remarks to the Author)

The authors have gone above and beyond in their reply and revisions. I didn't intend to make them do that much extra work.

I'm glad to see they have corrected the section on 3D imaging with AFM. The revisions are sufficient.

The NIL discussion was very thorough. I do still think it would not be that difficult for someone skilled in NIL to make 20 copies and then look at them with an SEM to find one with sufficient matching to the original pattern. The cloning though is irrelevant to the definition of PUF that they described where "unclonable" just means difficult to clone. This section is ok now.

I do still have problems with the section on fingerprints on rough surfaces. PS-PMMA is somewhat neutral in chi with air. The lamella in thick films often will template from the air-BCP interface. The BCP film is also not perfectly conformal and will likely be smoother for high frequencies than the underlying surface. The SEM is looking at the top surface. Seeing fingerprints on the top surface says nothing about the structure below the surface. There is no driving force to make lamella go all the way from the surface to the substrate with a perfect vertical orientation. Thick films could have multiple orientations under the surface or tilted lamella. Both of these cases will cause big challenges with transferring the fingerprint pattern to the substrate. Tilted lamella probably won't print. It isn't necessary for them to try the etch to transfer the BCP fingerprints to a rough surface. If it is easy to do, that would improve the manuscript. If they don't do the etch transfer, then they need to add a comment about how seeing the fingerprint on the BCP surface does not mean that it will actually be able to be transferred. Once they correct that section the manuscript will be ready to publish.

(Remarks on code availability)

Point-by-point response

We thank the Reviewers for the constructive and competent comments, which give us the opportunity to improve the overall quality of our manuscript. By providing new experimental results and new analyses, we provide here a point-by-point response addressing all the reviewer comments, in particular regarding the application of the pattern as a PUF and the stability of the pattern. Clearly, we have also extensively revised the main manuscript and supplementary information, accordingly. We believe that the revised version of our manuscript now complies with the high standard required for publication in *Nature Communications*.

Reviewer #1

In this paper, the authors reported disorder-based PUFs for anti-counterfeiting using nanopatterns formed by block copolymer self-assembly. The authors propose a user-independent strategy for successfully extracting nanopatterns from low-quality micrographs and show that nanopatterns can be encoded into binary code matrices that provide high bit uniformity and uniqueness. In addition, the authors also discussed the appropriate encoding of morphological nanopattern features, including defects and correlation lengths. Finally, the authors propose a computer vision-based strategy that enables robust authentication processes. Unfortunately, however, this study seems challenging in obtaining novelty in terms of block copolymer nanopattern-based PUF label formation and authentication using minutia, and significant improvement in achieving novelty is necessary. I expect this manuscript to become an outstanding paper after revision.

We thank the reviewer for the general positive assessment of our work and for comments that helped to improve the quality of our manuscript. According to reviewer suggestions, we have performed new experiments and we have clarified the novelty of our work in the revised manuscript, as detailed below.

C1. Several studies have reported using self-assembly of polymers to produce authentication tags. In this respect, the pattern manufacturing method and authentication elements shown in this study are similar to previous results. Therefore, if the paper's novelty is to increase accuracy through image processing, additional explanation is needed, especially in the abstract.

R1. We thank the reviewer for pointing out this aspect that was not properly highlighted through the previous manuscript. Main novelties of our work can be summarized as below.

- **We report for the first time on a novel strategy for directly engraving a nanoscale fingerprint-like pattern on a target material as an anticounterfeiting tag.** Besides showing that these artificial fingerprints can be engraved on a wide range of target materials, we also show in the revised manuscript new experimental results showing the feasibility of engraving an artificial fingerprint on a real object (refer to response R3).
- **We report on a new approach for extracting the genuine fingerprint-like pattern** from self-assembled nanostructures, that represents a fundamental aspect for properly accessing the PUF response without the introduction of artifacts and for providing noise robustness to the authentication process. In this context, a detailed comparison of our approach with state-of-the-art for pattern extraction is provided in the revised version of the manuscript.
- **We show that the encoding strategy of the nanopattern in a corresponding binary code matrix significantly affects security properties of the system.** In particular, we show that

the optimization of the encoding process is a crucial aspect for optimizing figures of merit of resulting binary code matrices, including bit uniformity, uniqueness, and unit entropy.

- **We propose a novel strategy for robust authentication and identification in real-world scenarios based on computer vision concepts.** Different from the standard approach based on encoding patterns in binary code matrices, this approach is shown to allow a robust authentication/identification of the nanopattern, by superimposing test images with images from a database through a homographic transformation. We show that this approach enables to avoid incorrect authentication/identification even in case of misalignment, translation, rotation and/or deformation of the end user image with respect to the database image.
- **We demonstrate a high stability of the nanopattern against aging and thermal stability,** an aspect often neglected, providing evidence of long-term reliable operation and robust authentication/identification in the range from -196 °C to 200 °C.

For these reasons, we believe that the novelty of our work complies with the high standard required for publication in *Nature Communications*.

Changes in the manuscript: according to the reviewer's remark, we have clarified the novelty of our work through the revised manuscript (mainly introduction and conclusions) and we have rewritten the abstract.

C2. Since the fingerprint tag produced in this study is characterized by nano size, expensive and space-limited equipment such as SEM is required to identify the morphology for authentication; therefore, to discuss actual application to the supply chain as shown in figure 1b, it will also be necessary to discuss difficulties that may arise in measurement or actual use.

R2. We agree with the reviewer's comment on the usually expensive instrumentation that is required to identify the morphology of the fingerprint tags for authentication/identification. However, there are several companies, nowadays, that provide numerous tabletop SEM instruments that are available in the market and that are more affordable than the conventional ones. The most recent tabletop SEMs can provide topographical information at magnifications of 10x up to 1,000,000x with a resolution up to 1 nm per pixel¹, which is more than sufficient to detect and analyze the fingerprint pattern through the fingerprint enhancement algorithm (that we demonstrate through the text that is a versatile algorithm for extracting nanopattern features even for poor-quality SEM images and that works with images from different SEM instruments). Furthermore, recent lab-based SEMs are able to handle very large and heavy objects (i.e. diameter 300 mm, height 130 mm and weight 8 kg) and enable multi-sample imaging and analysis^{2,3} (scanning areas of 100 cm x 100 cm with nanometric resolution) (we report an example in Figure R1 for the benefit of the reviewer). Anyway, the acquisition and identification of the fingerprint tag through SEM imaging inherently requires the insertion of the object inside a chamber and vacuum conditions, arising therefore difficulties in the measurement and in their actual use. In this context, alternative instruments such as AFM, can be employed to acquire the morphology with sufficient image quality and resolution of the fingerprint tag, that do not need the insertion of the object inside a space-limited chamber and can operate in air. Figure R8d and e show a representative AFM micrograph of an engraved fingerprint nanopattern and the relative *z*-profile, respectively. In this context, different techniques for accessing the nanopattern can be exploited depending on the object of interest.

The possibility of employing different measurement techniques such as SEM and AFM, expands the actual application of the proposed engraving technology in the supply chain. Notably, the measurand on AFM and SEM is different even when aiming at the same physical quantity due to their distinct measurement principles, i.e., the detection of interaction of the tip with the substrate and electron

beam-sample interaction, respectively (refer to the discussion reported in ref.⁴) Notably, by taking advantage of the complementary nature of AFM and SEM techniques, it is also possible to exploit some additional features of the engraved pattern as a second-level of authentication in a multiple-level security check. This includes the evaluation of the 3D structure of the pattern and z -profiles obtained by AFM measurements which is determined by the combination of BCP self-assembly and the subsequent engraving process and hardly cloneable through lithographic processes, therefore improving the authenticity of the pattern (detailed discussion in response R10).

[REDACTED]

Figure R1. Large chamber SEMs for large and heavy objects. From: ref² accessed on June 26th 2024.

Changes in the manuscript: The above discussion was included in the discussion paragraph of the revised manuscript to clarify the actual application of the proposed engraving technology (page 22 of the revised manuscript). AFM micrograph with relative z -profile of the engraved nanopatterns are added as new supplementary figures, Supplementary Figure S24d and e. Ref.^{3,4} have been included in the manuscript.

C3. The explanation in Figure 1f-i states that the self-assembly engraving technology can be applied directly to the surfaces of various products that are not flat, such as diamonds and cameras. However, it is technically very difficult to form the BCP self-assembled structure on an uneven surface and etch it uniformly. Therefore, explaining the technologies and problems required to apply engraving technology is necessary. Furthermore, regarding the SEM image inserted in Figure 1f-i, the figure may need to be modified as it may cause readers to misunderstand that the technology was applied on various product surfaces that are not flat.

R3. We thank the reviewer for pointing out this aspect. We agree that technical difficulties can be encountered on the formation of self-assembled BCP structures and the subsequent etching on uneven surfaces, such as curved^{5,6} and rough surfaces^{7,8}. In this context, we performed additional experiments to demonstrate the possibility of directly engraving the typical fingerprint pattern onto a surface that is both curved and rough. Specifically, we successfully formed BCP fingerprint patterns and engraving on an optical lens. Figure R2a and b shows the optical image of the glass optical lens and the SEM image of the uniform fingerprint pattern engraved on the lens' surface. The curvature of the optical lens was verified by optical profilometer measurements in Figure R2c and d, revealing a curvature of more than 400 μm on a selected area of 7.3 x 2.5 mm^2 . The assessment of the roughness of the optical lens made by optical profilometer measurements in Figure R2e and f revealed a highly

rough surface (RMS = 104.3 nm) that, however, did not impair the overall pattern formation and engraving process.

Regarding the reviewer's concerns on the etching uniformity, our results show that the proposed engraving technology is able to provide fingerprint patterns on uneven surfaces that retain a pattern quality sufficient to be detected and analyzed by the fingerprint enhancement algorithm. To this end, the developed fingerprint enhancement algorithm, able to extract the pattern even from low-quality SEM images as reported in Supplementary Figure S4, can successfully extract and binarize the nanopattern engraved on the curved optical lens, as shown in Figure R3.

Following the reviewer's recommendations, we have clarified in the caption of Figure 1f-i that the images of the objects are to be considered for demonstration purposes of the items in which the engraving process can be applied to. Indeed, based on the aforementioned results, in which we demonstrated the successful engraving of fingerprint patterns on curved and rough surfaces, we believe that the objects represented in Figure 1f-i can be considered as figurative for demonstration purposes of the items in which the engraving technology can be applied to. Anyway, in order to avoid misunderstanding, this was clearly specified in the figure caption.

Figure R2. Fingerprint patterns engraved on a curved and rough surface. **a.** Photograph of an optical glass lens and **b.** SEM image of the engraved fingerprint pattern. **c.** Optical profilometer measurement of the curvature of the optical lens on a $7.3 \times 2.5 \text{ mm}^2$ area and **d.** relative height profile with schematic representation of the lens measured in confocal imaging mode with a 5x optical lens showing the lens curvature. **e.** Optical profilometer measurement of the roughness of the optical lens on a $42 \times 35 \text{ }\mu\text{m}^2$ area with a 20x optical lens and **f.** the relative profile.

Figure R3. Binarization of a pattern engraved on a curved surface. Successful binarization of an SEM image of a fingerprint pattern engraved on a curved optical lens surface by exploiting the fingerprint enhancement algorithm. Scale bar is set to 500 nm.

Changes in the manuscript: To avoid any misunderstanding, we have clearly stated in the caption of Figure 1 a clarification that the images of the objects in Figure 1f-i are for demonstration purposes of the items in which the engraving technology can be applied to. We have stated in the manuscript (page 7) that the engraving technology can be applied also to curved and rough surfaces. New experimental results concerning the assessment of the engraving technology on uneven surfaces have been reported in Supplementary Figure S1, a sentence recalling these new data has been added at page 7. Assessment of the curvature and roughness of the optical glass lens has been performed by optical profilometer measurements and reported in Supplementary Figure S1c-f, a sentence recalling this has been added at page 7. Additional experimental results on the successful binarization of a nanopattern engraved on a curved surface has been reported in Supplementary Figure S6 and is discussed in the revised manuscript at page 11.

C4. In line 204th, the authors explain that the intrinsic randomness of the self-assembled structure can guarantee the uniqueness of the response of the PUF. However, the uniqueness of the response is expected to be affected by several factors, such as the deviation of the response depending on the characteristics or type of challenge input to the PUF and the averaging effect of the response depending on the size of the PUF label. Therefore, it seems necessary to further explain that the self-assembly's intrinsic randomness guarantees the response's randomness.

R4. While the overall morphology of block copolymer patterns (i.e. the lamellar shape with a given periodicity and long-range ordering) is determined by thermodynamic principles and remains reproducible, the detailed fingerprint-like patterns exhibit an element of randomness. This randomness is due to local fluctuations, kinetic factors, and defects that naturally arise during the self-assembly process.

The degree of randomness in these patterns is significantly influenced by the experimental conditions employed during the self-assembly, including parameters such as temperature, time, solvent, substrate, and thermal ramps.⁹ In our experiments, we employed the rapid thermal annealing (RTA) method.^{10,11} This technique allows for a high degree of control over the processing parameters and enables us to quench the kinetics of ordering into a specific timeframe with very rapid thermal ramps.

By utilizing the RTP method, we achieve precise control over the self-assembly conditions, thereby managing the extent of long-range ordering in the resulting patterns. This approach allows us to fine-tune the balance between thermodynamic ordering and kinetic trapping, providing a reproducible framework for studying the self-assembly of block copolymers while acknowledging the inherent variability in the detailed fingerprint-like structures.¹¹

Furthermore, a polydispersity index (PDI) different from 1 (in our experiments we used a BCP with PDI = 1.09) can be considered as another source of local randomness that enables the random formation of defects in the pattern.¹² Variations in the PDI lead to discrepancies in the size and distribution of the polymer blocks, contributing to local irregularities and imperfections in the final structure.

Furthermore, we have further explored the randomness of the binary matrices associated with nanopatterns through the NIST Statistical Test Suite 800-22 R1a and the analysis of autocorrelation function (refer to response R11 for details).

Changes in the manuscript: according to the reviewer's suggestion, we have further explained the self-assembly's intrinsic randomness through the revised manuscript in the introduction paragraph (page 6) and in the discussion paragraph (page 21). Ref¹² has been added in the manuscript. Furthermore, the results of the NIST SP 800-22 test on binary code matrices extracted from nanopatters are added in the manuscript in Supplementary Table 1 and 2, a sentence recalling this point has been added in the revised manuscript (page 14) and the reference of the NIST test (ref.¹³) has been added in the supplementary information. The analysis of the autocorrelation function is added in Supplementary Figure S10, a sentence recalling this aspect has been added in the revised manuscript (page 14).

C5. In Figure 2, the authors achieved improved image recognition capabilities through fingerprint enhancement, allowing different types of defects to be distinguished much better. However, in Figure 3, the key is generated only through the presence or absence of minutia and density. Therefore, discussing the strategies to actively utilize the advantages of improving image recognition ability through data processing in key generation, such as encryption of specific minutia types, will strengthen the authors' argument.

R5. As correctly mentioned by the reviewer, fingerprint enhancement allows for improved extraction of the correct pattern from images. This represents a fundamental aspect for exploiting these systems as image-based PUF, as detailed in the following.

Firstly, the extraction of the genuine pattern through fingerprint enhancement is crucial to identify only genuine defects avoiding bit flips in the resulting binary code matrices built from defects. Specifically, common binarization thresholding techniques lead to artifacts that consequently generate bit flips in the corresponding binary code matrices with respect to those obtained from the fingerprint enhancement. As analyzed in Figure R4, average median bit flips around 10% and outliers exceeding 60%, clearly indicate significant estimation of non-genuine defects when using different thresholding techniques when compared to fingerprint enhancement.

Secondly, fingerprint enhancement allows for the correct assessment of the defect type which is crucial when the defect types are considered for the generation of analog code matrices. Different thresholding techniques, when compared to fingerprint enhancement, generate artifacts or underestimate specific defects that eventually lead to errors in the encryption of specific minutiae. As for example reported in Figure R5, errors in the detection of terminal dots defects can be noticed through Mean thresholding binarization, whereas overestimation of defects and detection of additional 4-way junctions are obtained through Median thresholding binarization.

Figure R4. Bit flips in binary code matrices obtained through different binarization methods with respect to fingerprint enhancement. Box plots representing the percentage of bit flips observed in binary code matrices when the nanopattern is binarized with conventional thresholding methods against binarization from the fingerprint enhancement algorithm. Midline represents median value, boxes the 25th and 75th percentiles, whiskers the 1.5 IQR (interquartile range), and black squares the outliers.

Figure R5. Binary code matrices comparison. Direct comparison of analog binary code matrices generated from different binarization thresholding methods of the same SEM image. The pixel is blue in presence of positive phase dot defects; pixel is red in presence of positive phase terminal point defects; pixel is yellow in presence of positive phase 3-way junction defects; pixel is olive green in presence of positive phase 4-way junction defects; pixel is light green in presence of positive phase dot and positive phase terminal point defects; pixel is orange in presence of positive phase terminal point and positive phase 3-way junction defects; the pixel is light blue in presence of positive phase dot defects and positive phase 3-way junction defects; the pixel is dark green in presence of positive phase 3-way junction defects and positive phase 4-way junction defects; pixel is violet in presence of positive phase dot defects, positive phase terminal point and positive phase 3-way junction defects; pixel is brown in presence of positive phase dots, positive phase 3-way junction and positive phase 4-way junction defects.

Changes in the manuscript: The above discussion on the correct extraction of the genuine pattern through fingerprint enhancement compared to other thresholding methods has been included in the manuscript (page 15), new analyses here reported in Figure R4 and Figure R5 has been added as new Supplementary Figures S14 and S15.

C6. The thermal stability discussed in Figure 4e will vary greatly depending on the constituent materials of the product to be engraved or heat treatment conditions. For example, if a fingerprint is engraved on a product made of ceramic, it will be thermally stable even in a high-temperature environment of 300 degrees, but in the case of a polymer, it is questionable whether it will be thermally stable under the same conditions. In this regard, explaining the importance of material selection to ensure thermal stability and the temperature limit that can maintain thermal stability would be better.

R6. As correctly mentioned by the reviewer, the thermal stability of the pattern relies on the intrinsic properties of the target material to be engraved. Therefore, the importance of the material selection is crucial for the specific application in order to ensure stability in the working environment, such as high and low temperature conditions. The proper choice of the material needs to be considered in function of the working application of the object to prevent degradation of the fingerprint pattern and to avoid the impairment of the authentication/identification process, as well as on the target material/object.

Depending on the final application/object of interest, it is worth mentioning that an alternative solution for realizing highly stable nanopatterns is represented by the sequential infiltration synthesis (SIS) of the polymeric BCP structures consisting in replicating (without engraving) the pattern with refractory oxides (such as Al_2O_3 and TiO_2).¹⁴ This process relies on the selective infiltration and growth of metal oxides inside one of the constituent blocks of self-assembled BCPs, carried out in an atomic layer deposition apparatus. The subsequent polymer removal reveals a nanostructured metal oxide that perfectly replicates the pattern morphology of the self-assembled BCP, which is in this case the fingerprint pattern. Examples of Al_2O_3 and TiO_2 nanopattern realized through SIS technique are reported in Figure R6. This approach can ensure the realization of nanopatterns that are stable at high temperature (almost 2273 K), with temperature stability that can be even higher than the object where the pattern is applied.

Concerning thermal stability, we complemented with new results the previous version of the manuscript, showing that nanopatterns engraved on the SiO_2 substrate are stable also at cryogenic conditions (refer to response R12 for details on new experimental data). Thus, our results demonstrate PUF stability in a large temperature range, from $-196\text{ }^\circ\text{C}$ to $200\text{ }^\circ\text{C}$.

Figure R6. Metal oxide replica of fingerprint patterns. SEM image of fingerprint patterns replicated with **a.** Al_2O_3 and **b.** TiO_2 by sequential infiltration synthesis. Scale bar is set to 500 nm.

Changes in the manuscript: The above discussion was included in the discussion paragraph of the revised manuscript to clarify the importance of the material selection (page 24). Supplementary Figure S25 showing examples of fingerprint patterns replicated with Al_2O_3 and TiO_2 by sequential infiltration synthesis is added in Supplementary Information. Ref.¹⁴ has been added in the manuscript.

C7. Strictly speaking, the concept of PUF is an object that performs authentication with a pair of inputs called 'challenge' and an output called 'response' without exposing the internal structure. In this respect, the fingerprint pattern fabricated by the authors is close to an anti-counterfeit tag or identifier because authentication must be performed by verifying the internal structure. Therefore, if the authors briefly mention that the fingerprint pattern is close to the anti-counterfeit identifier, the author will be able to prevent confusion among readers.

R7. We thank the reviewer for pointing out this aspect. Our nanoscale artificial fingerprints can be considered as image-based PUFs, a class of PUF based on random visual features of the physical object (as discussed in Ref¹⁵) that can be exploited, as correctly pointed out by the reviewer, as anti-counterfeit identifiers.

Changes in the manuscript: As suggested by the reviewer, a statement recalling that the fingerprint pattern is close to an anti-counterfeit tag has been added in the abstract, introduction, results and conclusion paragraphs (pages 4, 12 and 25). It was clarified through the text that artificial fingerprints can be considered as image-based PUFs (page 12). Ref.¹⁵ has been added in the manuscript.

C8. It would be better to write the full name before using the abbreviated form in the text. For example, 'RCP' in the method part is supposed to mean random copolymer, but it may need to be clarified for those unfamiliar with the content. Reaffirming the format and typos of the paper will help readers understand. For example, in line 150th, there is no period, and in line 422, 'counterfeiting' should be 'anti-counterfeiting.'

R8. We thank the reviewer for pointing out the mistakes and typos that helped us to improve the readability of the overall manuscript.

Changes in the manuscript: The full name random copolymer has been added in the methods section before the abbreviation RCP (page 26). All the manuscript was revised to remove typos and format errors.

Reviewer #2

Murataj et al describe a study of using fingerprint patterns from block copolymers as nanoscale physically unclonable functions (PUFs) for authentication. The idea is very interesting. Block copolymers provide a pretty large range of random structures. The manuscript does a very thorough job of analyzing images of the fingerprint patterns and developing a robust method for a matching algorithm and analysis of the entropy of the structures/defects in block copolymer patterns. They also transferred patterns to a variety of substrates and evaluated the physical robustness of the patterns over time and temperature. There are two main limitations to the work. The verification process is very difficult (requires a high-resolution scanning electron microscope (SEM) with the nanopattern accessible) and they do not discuss if the nanopatterns are in-fact unclonable (I do not think they are). The block copolymer structure as described is more like a high entropy nanoscale QR code than a PUF. I discuss these two topics in more detail below.

We thank the reviewer for the general positive assessment of our work and for constructive comments that helped to improve our work.

C9. Verification/reading of the PUF function requires a scanning electron microscope to image the nanopattern. The manuscript describes the difficulty of authenticating the PUF as an enhancement to the security. I consider it to be a severe limitation to the use of the PUF. It would work much better and be more usable if there was an easier way to authenticate the PUF function. Putting your chip into an SEM is pretty involved, especially after the chip has been packaged. In that case you would have to open up the package and destroy the chip. As described, it is only really readable during wafer fabrication. In general there is pretty good security during the wafer fabrication process and the main counterfeiting concern is during packaging and post-packaging. It would be much better if there was an electrical or optical method for measuring the nanopattern that would get out a unique signature from the PUF. An electrical or optical method might also be more sensitive to nanoscale details of the nanopattern than the SEM imaging and defect analysis is. It also could be sensitive to the materials and the nanoscale 3D structure of the nanopattern. These additional factors would make it much more difficult to clone the PUF.

R9. We thank the reviewer for pointing out these aspects. We agree with the reviewer that conventional SEMs are usually capable of handling only limited sample sizes, which could be seen as a limitation. However, there are nowadays numerous SEM instruments available in the market equipped with large chambers (up to few meters) capable of accommodating very large samples without the need for cutting.^{16,17} These advanced SEMs are mainly used in the field of manufacturing and quality control, archeology, geology, forensic and materials sciences, and are already installed in many logistics centers, universities, and research institutes, facilitating easier access and usability.^{2,3,18}

Furthermore, the most recent lab-based SEM are capable of handling large samples in terms of dimensions (i.e. diameter 300 mm, height 130 mm weight 8 kg) and number (scanning areas of 100 cm x 100 cm with nanometric resolution) (example reported in Figure R7 for the benefit of the reviewer). In particular, there are tabletop SEM instruments available in the that are more affordable than the conventional ones still providing information at magnifications of 10x up to 1,000,000x with a resolution up to 1 nm per pixel¹ (refer to response R2 for details).

In this context, alternative instruments such as AFM, can be employed to acquire the morphology with sufficient image quality and resolution of the fingerprint tag, that do not need the insertion of the object inside a space-limited chamber and can operate in air (refer to comment R10 for a detailed discussion and for an example of nanopattern extracted through AFM). In this context, different techniques for accessing the nanopattern can be exploited depending on the object of interest and

target application. The possibility of employing different measurement techniques such as SEM and AFM, expands the actual application of the proposed engraving technology in the supply chain. Notably, the measurand on AFM and SEM is different even when aiming at the same physical quantity due to their distinct measurement principles, i.e., the detection of interaction of the tip with the substrate and electron beam-sample interaction, respectively (refer to the discussion reported in ref.⁴).

[REDACTED]

Figure R7. Large chamber SEMs for large and heavy objects. From: ref² accessed on June 26th 2024.

In this context, it is important to remark that the nanopattern can be accessed also through AFM that provides a viable alternative to SEM imaging. In this case, the nanopattern can be acquired in air and without the need of inserting the object into a chamber.

Our nanoscale artificial fingerprints can be considered as image-based PUFs, a class of PUF based on random visual features of the physical object that can mainly be targeted to anti-counterfeit applications since the input is represented by a fixed challenge (refer to the discussion on these aspects reported in ref.¹⁵). In perspective, image-based PUF based on artificial fingerprints can be integrated with optical and/or electrical readout to generate a larger amount of unpredictable challenge/response pair (similarly to the approach reported in ref.¹⁹), as required for remote authentication applications and secret key generation.

It is worth noticing that the proposed artificial fingerprints can be exploited not only for chips but for a wide range of materials/objects. In this context, the high versatility of the block copolymer self-assembly allows extending the described process to several materials, including metals, ceramics and polymers. Notably, the PUF is conceived to be engraved on the object in a place where can be easily accessed, so there is no need of destroying the object itself for authentication/identification.

Changes in the manuscript: The above discussion was included in the discussion paragraph of the revised manuscript (page 23). Ref.³ has been added in the manuscript.

C10. The proposed structures are not actually unclonable as described. If the nanopattern is available for SEM imaging, it could be cloned. The manuscript only addresses this concern by saying that the PUF could be hidden on the chip. This isn't really a good security feature because the location is difficult to protect and could be discoverable if the PUF is widely used. There are two methods I'm aware of that could clone the proposed PUFs. The first method is nanoimprint lithography (NIL). NIL

can be used to make a replica of the PUF nanopattern and transfer it to a new chip (or many new chips). Their verification/reading method using imaging SEM is not sensitive to the depth of the pattern or the material that the pattern is made out of. They are simply analyzing the 2D binarized fingerprint structure and defects on the 20-50 nm length scales. These defects could easily be replicated with the nm-scale resolution of NIL. The second method is e-beam lithography. Since the PUF nanostructure is imageable, it would be possible to use the digital image to create a pattern for e-beam lithography. The 2D pattern with sub-20 nm resolution would require a very good e-beam tool and some process development but would be doable if the authentication is simply through comparison of large defects in the fingerprint pattern. To make the nanopattern unclonable, it needs to be buried within the chip structure to protect it from imaging, and a non-imaging method of verification/reading would need to be developed. If an electrical/optical method was sensitive to nanoscale fluctuations in the 3D nanostructure it would be effectively unclonable since these features are much more difficult to fabricate controllably even if the PUF was imageable.

R10. We thank the reviewer for pointing out this important aspect. The vast majority of image-based PUF reported in literature relies on macro/microscale patterns. This includes, for example, the development of physical unclonable functions based on random fractals with microscale features that can be identified through optical microscopy²⁰ and on randomly positioned micrometric features exploited for authentication via binary keys derived from optical images.²¹ Similarly, other works generate cryptographic keys for PUFs from micrometric topographical spots obtained by dielectric breakdown and detected by optical means.¹⁵ While these works rely on visual features of the physical objects at the macro/microscale that can be potentially cloned through lithographic techniques, our work relies on nanoscale features that are hard to be physically cloned (note that we consider unclonability on the physical level only, while not considering mathematical clones).²²

On one side, the physical unclonability is guaranteed by the randomness driving the self-assembly process, as demonstrated by evaluating the inter-HD of binary code matrices obtained by considering different nanopatterns (refer to Figure 3g) that are close to the ideal value of 0.5.

On the other side, other nanofabrication processes can be exploited by a malicious party (counterfeiter) for attempting to clone the nanopattern. Even if it is in principle possible to reproduce a pattern whose 2D binarized fingerprint matches with the original pattern to be cloned with EBL or nanoimprinting lithography, it would require very large efforts and the cost for clonability is expected to be almost prohibitive. In any case, there are some features of the nanostructures that cannot be reproduced through these techniques and that can be used as a second level of authentication in a multiple-level security check scenario. This includes the evaluation of Line Edge Roughness (LER) and Line Width Roughness (LWR) of the lamellar pattern (Figure R8), that are intrinsic properties of the BCP self-assembly process²² and are hardly cloneable through lithographic processes. Figure R8b and c reports a detailed view of the lamellar structure and corresponding binarized pattern (with Otsu binarization), respectively. Here, LER and LWR are clearly visible (values of LER and LWR extracted from Figure R8b and evaluated through ADA block²³ are 1.39 nm and 2.35 nm, respectively). A further additional level of authentication can be represented by the analysis of the 3D structure of the pattern and z -profiles through AFM, as reported in Figure R8d and e, respectively. The 3D structure of the pattern, which relies on the combination of BCP self-assembly and the subsequent engraving process, is hardly reproducible through lithographic techniques.

For all these reasons, as pointed out also by reviewer 3, we believe that the artificial fingerprint engraved through block-copolymers can be exploited as image-based PUF where the authenticity of the pattern can be probed with multiple levels of authentications if, besides the 2D binarized fingerprint structure, additional parameters such as LER, LWR and 3D morphology are stored in the database of the manufacturer.

Figure R8. Additional morphological parameters. **a.** Schematic of line edge roughness (LER) and line width roughness (LWR). **b.** SEM image of a fingerprint pattern and **c.** the binarized image by ADA block with Otsu binarization. Scale bar is set to 100 nm. The LER and LWR calculated on binarized image of the engraved nanopattern reported in panel **c.** are 1.39 nm and 2.35 nm respectively. **d.** 3D morphology of a fingerprint pattern measured by AFM in tapping mode over a $400 \times 800 \text{ nm}^2$ and **e.** example of a z-profile extracted from the AFM image.

Changes in the manuscript: The above discussion was included in the discussion paragraph of the revised manuscript to discuss the physical unclonability of our nanopatterns. New experiments showing the LER, LWR and AFM micrograph with relative z-profile of the engraved nanopatterns, here reported as Figure R8a-c and R8d-e are added in Supplementary Figure S24. Ref.²² has been added in the manuscript.

Reviewer #3

This paper presents a novel physically unclonable function (PUF) manufacturing technique that exploits block-copolymers (BCPs) to form unique nanopatterns and precisely copy these patterns onto multiple substrate materials through pattern transfer techniques. This process not only takes advantage of the inherent randomness of nanoscale material self-assembly to generate complex nanostructures that are difficult to replicate or clone, but also, through the developed algorithm, encodes these nanopatterns into binary code matrices with high entropy and high uniqueness, which greatly increases the coding density and enables large amounts of information to be stored in a very small area. In addition, the authors propose a computer vision-based authentication and identification strategy to further enhance the PUF's security and feasibility. The experimental results validate the high reliability of the fabricated PUF, under aging and high temperature conditions. Features extracted from scanning electron microscope (SEM) images through computer vision technology were successfully converted to binary codes that are more suitable for security authentication. Several detailed comments are provided as follows:

We thank the reviewer for the general positive assessment of our work and for comments that helped to improve the quality of our manuscript including also new experiments.

C11. While the paper discusses the inherent stochasticity of self-assembled block copolymers, it would be necessary to include a more comprehensive analysis of the randomness of the nanopatterns and PUF. It is expected to add more test results to further verify the PUF's randomness, such as autocorrelation function (ACF) test and NIST SP 800-22 test results.

R11. We thank the reviewer for pointing out this aspect. According to reviewer suggestions, we have performed the analysis of the autocorrelation function and the suggested NIST SP 800-22 test to include a more comprehensive analysis of the binary code matrices randomness, as discussed below.

Results of the NIST SP800-22 test¹³ performed on the dataset of two-hand binary code matrices extracted from nanopatterns are reported in Table R1, while Table R2 reports the parameter values used for the test. Note that those tests have been performed over binary code matrices (10 x 10 pixels) with pixel size of $238 \times 238 \text{ nm}^2$, i.e., when bit uniformity is ~ 0.5 , the HD between binary code matrices of ~ 0.5 and bit entropy is close to 1, further verifying the possibility of generating binary code matrices with high randomness. As can be observed from Table R1, given the proportion results of sequences passing a test (p -value ≥ 0.01), there is evidence that the binary code matrices are random for all the statistical tests applicable. While there is a non-uniform distribution of p -values of binary code matrices for the Frequency (Monobit) Test, for all the other applicable ones the binary code matrices can be considered to be uniformly distributed. Based on those results, the binary code matrices pass most of the applicable tests of the NIST test suite.

Concerning the autocorrelation function (ACF), we report as an example of the ACF as a function of the lag for six different binary code matrices, extracted from binarized fingerprint patterns, organized as an array of 100 elements by placing the rows of the matrix one after the other as reported in Figure R9. As can be observed, autocorrelation values for lag > 0 lays mainly within the boundary of the 95% confidence (whereas only few values are outside the confidence band) indicating that there is negligible correlation between pixels (note that the autocorrelation value of 1 at lag = 0 corresponds to the correlation of the pixel against itself).

Table R1: Results of NIST SP 800-22 statistical test suite.

NIST statistical test ^a	p -value _T	Proportion $\alpha = 0.01$ [Pass/Total]	Proportion Results ^b
Frequency (Monobit) Test	0.000014*	198/200	Pass
Frequency Test within a Block	0.616305	197/200	Pass
Cumulative Sums (Cusum) Test	0.062821	197/200	Pass
	0.001824	196/200	Pass
Runs Test	0.181557	198/200	Pass
Approximate Entropy Test	0.002758	198/200	Pass
Serial Test	0.419021	196/200	Pass
	0.989786	198/200	Pass
^a200 different binary code matrices of 100 bits each were used to perform all possible NIST tests given our data. ^b The minimum pass rate for each statistical test is approximately = 193 for a sample size = 200 binary sequences. *Non-uniform distribution of binary code matrices since p-value_T < α, with significance level $\alpha = 0.0001$.			

Table R2: Parameters adopted for the NIST test.

NIST statistical test parameter	Value
Access (n):	100
Bitstreams:	200
Block Frequency Test - block length (M):	20
Approximate Entropy Test - block length (m):	1
Serial Test - block length (m):	4

Figure R9. Autocorrelation function as a function of lags. Examples of autocorrelation evaluated on vectors with 100 elements realized by placing rows of the 10x10 matrices one after the other. The autocorrelation as a function of lags is reported as an example for six different nanopatterns.

Changes in the manuscript: The results of the NIST SP 800-22 test are added in the manuscript in Supplementary Table 1 and 2, a sentence recalling this point has been added in the revised manuscript (page 14) and the reference of the NIST test (ref.¹³) has been added in the supplementary information. The analysis of the autocorrelation function is added in Supplementary Figure S10, a sentence recalling this aspect has been added in the revised manuscript (page 14).

C12. How is the reliability performance for the fabricated PUF under the low temperature, such as the range of $-40^{\circ}\text{C}\sim 0^{\circ}\text{C}$?

R12. According to the reviewer's request, we assessed the reliability performance of the fabricated PUF under low temperatures, in a temperature range well below -40°C . Specifically, the PUF were thermally treated at -196°C for 30 minutes. Figure R10a shows the heatmap matrix representing the fractional HD between images from the test set and after thermal annealing of 100 PUF devices at -196°C for 30 min, and Figure R10b shows the corresponding intra and inter distributions. The mean values and std. dev. of intra and inter-HD distributions are 0.17 ± 0.04 and 0.499 ± 0.003 , respectively. Also in this case, successful authentication/identification of all PUF patterns without any false positive or false negative have been achieved by exploiting the decision threshold of 0.45 previously evaluated through the comparison of the test set of images with the database.

These new results further highlight the high thermal stability of PUF devices in a large temperature range, from -196°C to 200°C .

R10. Authentication/Identification through images acquired after low temperature thermal treatment. **a.** Heat-map matrix representing the fractional HD between images from the test set and after thermal annealing of 100 PUF devices at -196°C for 30 min, and **b.** the corresponding intra and inter distributions. Dashed line in intra and inter distributions represent the decision threshold evaluated from the comparison between the test set and the database.

Changes in the manuscript: The above results have been discussed in the results paragraph of the revised manuscript (page 19), to prove the reliability of the PUF also at low temperatures. The heatmap matrix of the fractional HD and the intra and inter-HD distributions are added in Figure 4. Mean values and std. dev. of intra and inter-HD distributions were added as new items in the revised Supplementary Table 3. Experimental details on the low temperature treatment have been added in the method paragraph.

C13. There is no direct comparison with other existing materials or techniques applied for PUF, please make some comparison with the state-of-the-PUF designs in terms of the widely-accepted figures of merit, such as reliability, randomness, uniqueness, normalized bitcell area, etc.

R13. We agree with the reviewer that the text should be improved to appropriately position our research results in the state-of-the art of related activities. To this end, we collected widely accepted figures of merit from the recent literature in order to make a direct comparison of our work with other existing materials and techniques for the realization of PUF. Among the PUF performance parameters we listed in the table below the number of devices that constitute the database set, randomness in terms of entropy (or approximate entropy from NIST test), uniqueness in terms of Hamming distance, uniformity, area of one normalized bitcell, encoding capacity, aging and temperature stability. As it is possible to observe from Table R3, we provide in our work a complete set of figure-of-merits as on the best of our knowledge well as we have provided a complete characterization of the PUF stability over time and in temperature, an essential aspect for the reliability of PUF devices sometimes neglected in other works.

Changes in the manuscript: A table listing a direct comparison of our work with the recent literature has been added as a new supplementary table (Supplementary Table 4). A sentence recalling the table has been added in the main manuscript, page 21. New ref.^{25,27,28,30–32,35–37} have been added in the supplementary information file.

Table R3. Direct comparison of our work with other existing materials and techniques applied for PUF. The table below reports a comparison of figures-of-merit of various PUF devices, including the dataset size, randomness in terms of unit entropy (or approximate entropy from NIST test), uniqueness in terms of Hamming distance, uniformity, area of one normalized bitcell, encoding capacity, aging and temperature stability in comparison to the literature. ^aThe randomness of binary-bit sequences were tested by statistical National Institute of Standards and Technology (NIST) test suite (NIST SP 800-22).¹³ *Values referred to PUF topography microstructures. [#]Pass rate not specified. [§]Values referred to image-based nanoidentifiers.

Ref	Database set	Approximate Entropy Test [Pass/Total]	Randomness (entropy)	Uniqueness (inter-HD)	Uniformity (bit uniformity)	Normalized bitcell area	Encoding capacity	Aging stability	Temperature stability
Our work^a	200	198/200	0.99 ± 0.01	0.50 ± 0.06	0.52 ± 0.06	2.38 × 2.38 μm²	1.3 × 10³⁰	6 months	-196 – 200 °C / 30 min
Ref ^{21a}	30	59/60	/	0.495 ± 0.033	0.495	/	2 ²²⁷	Not tested	1000 °C / 1h
Ref ²⁰	1100	/	/	/	/	/	10 ³⁴⁸	6 months	-40 – 750 °C
Ref ²⁴	100	/	/	0.497 ± 0.013*	0.494 ± 0.005*	950 × 950 μm ²	~ 10 ³¹⁷⁹⁶	2 months [§]	Not tested
Ref ²⁵	90	/	0.9714 ± 0.0416	0.4863 ± 0.0800	51.96 ± 0.19%	204 × 204 μm ²	~ 10 ⁷⁶³	7 days	200 °C
Ref ²⁶	15	/	~ 1	~ 0.5	~ 0.5	100 × 100 μm ²	~ 2 ³⁸⁰⁰⁰⁰	Not tested	Not tested
Ref ^{27a}	80	77/80	/	0.4997 ± 0.0363	0.4900 ± 0.0367	/	~ 2 ¹⁹⁰	Not tested	Not tested
Ref ²⁸	100	/	/	0.5000	0.4996	100 × 100 μm ²	/	Not tested	400 °C / 24h
Ref ²⁹	9	/	/	49.72 ± 3.37%	50.40 ± 1.67%	0.139 × 0.139 μm ²	/	1.5 months	Not tested
Ref ^{30a}	30	60/60	Red 60/60 Green 60/60 Blue 58/60	Red 0.506 ± 0.060 Green 0.507 ± 0.066 Blue 0.505 ± 0.062	Red 0.499 ± 0.055 Green 0.504 ± 0.052 Blue 0.507 ± 0.062	/	~ 4 ⁹⁵	/	70 °C / 10 days
Ref ¹⁹	300 [§]	/	1.080 ± 0.127 [§]	0.5007 ± 0.0545 [§]	/	0.75 × 0.75 μm ²	2.83 × 10 ^{163§}	Not tested	150 °C / 1h
Ref ^{31a}	100	Passed [#]	/	0.38	0.505	10 × 10 mm ²	10 ²⁵⁰⁰	11 days	45 °C / 11 days
Ref ^{32a}	100	86%	/	Ceramic Table Micro 0.492 ± 1.8 × 10 ⁻⁵ Ceramic Portable Micro 0.484 ± 4.7 × 10 ⁻⁵ Metal Table Micro 0.499 ± 1.4 × 10 ⁻⁵ Metal Portable Micro 0.494 ± 1.7 × 10 ⁻⁵	/	/	2 ²⁸⁷³⁵	/	/
Ref ³³	24	/	~ 1	0.47	~ 1	/	2 ⁶⁴	5 days	-175 – 105 °C
Ref ³⁴	100	/	/	0.494 ± 0.0056	/	200 × 200 μm ²	2 ¹¹⁸⁴⁸	Not tested	380 °C / 30 min
Ref ^{35a}	30	60/60	/	0.5032 ± 0.0458	0.5	7 × 7 mm ²	2 ¹²⁰	60 days	Not tested
Ref ³⁶	15	/	x _{axis} = 0.93 ± 0.06 y _{axis} = 0.91 ± 0.05	~ 13	/	640 × 640 μm ²	10 ¹²³³	Not tested	Not tested
Ref ³⁷	100	/	~ 1	~ 0.5	/	3.5 × 3.5 mm ²	2 ¹⁷⁵⁰	Not tested	Not tested

References

1. <https://www.jeolusa.com/NEWS-EVENTS/Blog/choosing-the-right-scanning-electron-microscope-for-your-laboratory>, accessed on 18th July 2024.
2. <https://www.jeolusa.com/RESOURCES/Electron-Optics/Documents-Downloads/large-direct-access-chamber-sems>, accessed on 26th June 2024.
3. Aretz, A. *et al.* In situ investigation of production processes in a large chamber scanning electron microscope. *Ultramicroscopy* **193**, 151–158 (2018).
4. Golden, T. W. & Murphy, R. D. Report of Investigation. in *A Guide to Forensic Accounting Investigation* vol. 8011 363–387 (Wiley, 2012).
5. Jin, H. M. *et al.* Laser Writing Block Copolymer Self-Assembly on Graphene Light-Absorbing Layer. *ACS Nano* **10**, 3435–3442 (2016).
6. Cho, J. H. *et al.* Ultrasooth Polydopamine Modified Surfaces for Block Copolymer Nanopatterning on Flexible Substrates. *ACS Appl Mater Interfaces* **8**, 7456–7463 (2016).
7. Sivaniah, E. *et al.* Symmetric Diblock Copolymer Thin Films on Rough Substrates. Kinetics and Structure Formation in Pure Block Copolymer Thin Films. *Macromolecules* **38**, 1837–1849 (2005).
8. Man, X., Tang, J., Zhou, P., Yan, D. & Andelman, D. Lamellar Diblock Copolymers on Rough Substrates: Self-Consistent Field Theory Studies. *Macromolecules* **48**, 7689–7697 (2015).
9. Bates, F. S. & Fredrickson, G. H. Block copolymer thermodynamics: Theory and experiment. *Annu Rev Phys Chem* **41**, 525–557 (1990).
10. Ferrarese Lupi, F. *et al.* Rapid thermal processing of self-assembling block copolymer thin films. *Nanotechnology* **24**, 315601 (2013).
11. Ceresoli, M. *et al.* Evolution of lateral ordering in symmetric block copolymer thin films upon rapid thermal processing. *Nanotechnology* **25**, 275601 (2014).
12. Lynd, N. A., Meuler, A. J. & Hillmyer, M. A. Polydispersity and block copolymer self-assembly. *Prog Polym Sci* **33**, 875–893 (2008).
13. Bassham, L. E. *et al.* *A Statistical Test Suite for Random and Pseudorandom Number Generators for Cryptographic Applications*. <https://nvlpubs.nist.gov/nistpubs/Legacy/SP/nistspecialpublication800-22r1a.pdf> (2010) doi:10.6028/NIST.SP.800-22r1a.
14. Cara, E. *et al.* Recent advances in sequential infiltration synthesis (SIS) of block copolymers (BCPs). *Nanomaterials* **11**, 1–26 (2021).
15. Shariati, S., Standaert, F.-X., Jacques, L. & Macq, B. Analysis and experimental evaluation of image-based PUFs. *J Cryptogr Eng* **2**, 189–206 (2012).
16. <https://www.nature.com/articles/d42473-019-00202-8>, accessed on on June 26th 2024.
17. <https://www.tescan.com/tescan-s8000-a-versatile-microscope-with-a-variety-of-imaging-modes/>, accessed on June 26th 2024.

18. <https://www.zeiss.com/microscopy/en/products/sem-fib-sem/sem/multisem.html>, accessed on June 26th 2024.
19. Kim, J. H. *et al.* Nanoscale physical unclonable function labels based on block copolymer self-assembly. *Nat Electron* **5**, 433–442 (2022).
20. Sun, N. *et al.* Random fractal-enabled physical unclonable functions with dynamic AI authentication. *Nat Commun* **14**, 2185 (2023).
21. Esidir, A., Pekdemir, S., Kalay, M. & Onses, M. S. Ultradurable Embedded Physically Unclonable Functions. *ACS Appl Mater Interfaces* (2024) doi:10.1021/acsami.4c01726.
22. Srinivas, G., Swope, W. C. & Pitera, J. W. Interfacial Fluctuations of Block Copolymers: A Coarse-Grain Molecular Dynamics Simulation Study. *J Phys Chem B* **111**, 13734–13742 (2007).
23. Murphy, J. N., Harris, K. D. & Buriak, J. M. Automated Defect and Correlation Length Analysis of Block Copolymer Thin Film Nanopatterns. *PLoS One* **10**, e0133088 (2015).
24. Zhang, J. *et al.* An all-in-one nanoprinting approach for the synthesis of a nanofilm library for unclonable anti-counterfeiting applications. *Nat Nanotechnol* **18**, 1027–1035 (2023).
25. Meijs, Z. C. *et al.* Pixelated Physical Unclonable Functions through Capillarity-Assisted Particle Assembly. *ACS Appl Mater Interfaces* (2023) doi:10.1021/acsami.3c09386.
26. Chen, P. *et al.* Programmable Physical Unclonable Functions Using Randomly Anisotropic Two-Dimensional Flakes. *ACS Nano* **17**, 23989–23997 (2023).
27. Kim, M. S. & Lee, G. J. Visually Hidden, Self-Assembled Porous Polymers for Optical Physically Unclonable Functions. *ACS Appl Mater Interfaces* **15**, 4477–4486 (2023).
28. Zhang, T. *et al.* Multimodal dynamic and unclonable anti-counterfeiting using robust diamond microparticles on heterogeneous substrate. *Nat Commun* **14**, (2023).
29. Porti, M., Redón, M., Muñoz, J., Nafría, M. & Miranda, E. Oxide Breakdown Spot Spatial Patterns as Fingerprints for Optical Physical Unclonable Functions. *IEEE Electron Device Letters* **44**, 1600–1603 (2023).
30. Kiremitler, N. B. *et al.* Tattoo-Like Multi-Color Physically Unclonable Functions. *Adv Opt Mater* (2023) doi:10.1002/adom.202302464.
31. Wu, J. *et al.* A High-Security mutual authentication system based on structural color-based physical unclonable functions labels. *Chemical Engineering Journal* **439**, (2022).
32. Li, Q. *et al.* Intrinsic Random Optical Features of the Electronic Packages as Physical Unclonable Functions for Internet of Things Security. *Adv Photonics Res* **3**, (2022).
33. Dodda, A. *et al.* Graphene-based physically unclonable functions that are reconfigurable and resilient to machine learning attacks. *Nat Electron* **4**, 364–374 (2021).
34. Li, Q. *et al.* Physical Unclonable Anticounterfeiting Electrodes Enabled by Spontaneously Formed Plasmonic Core–Shell Nanoparticles for Traceable Electronics. *Adv Funct Mater* **31**, (2021).
35. Leem, J. W. *et al.* Edible unclonable functions. *Nat Commun* **11**, 328 (2020).

36. Wali, A. *et al.* Biological physically unclonable function. *Commun Phys* **2**, 39 (2019).
37. Nocentini, S., Rührmair, U., Barni, M., Wiersma, D. S. & Riboli, F. All-optical multilevel physical unclonable functions. *Nat Mater* **23**, 369–376 (2024).

Point-by-point response

We thank Reviewer 1 and Reviewer 3 for the positive assessment of our revised manuscript, as well as Reviewer 2 for constructive and competent comments that give us the possibility to further improve the quality of our manuscript. Addressing all Reviewer 2 comments, we have revised the manuscript and supplementary information (changes of this second revision are highlighted in blue). We believe that this revised version of our manuscript now complies with the high standard required for publication in *Nature Communications*.

Reviewer #1

I appreciate the quick work the authors did in addressing my concerns with the first manuscript and the effort the authors made to address each of my concerns in their response to my initial review. The work presented here is truly impressive. In conclusion, I have no further suggestions for improvement.

We thank the reviewer for appreciating our work and for the positive assessment of our work.

Reviewer #2

The authors have revised the manuscript and added additional data and discussion. They have added additional more detailed information on the entropy of the structure and measurement repeatability and completed a detailed statistical analysis based on a NIST standard. This work has improved the manuscript a lot.

We thank the Reviewer for the positive evaluation of the changes made in the revised version of the manuscript.

C1. The fundamental remaining problem with the manuscript is the combined difficulty of fabrication and imaging/verification of the nanostructure. Putting an object in an SEM is not an easy way to verify the PUF. It would be much better if they had an easier method like optical scatterometry (OCD).

R1. OCD scatterometry is a highly effective technique for measuring periodic and well-ordered structures, such as those found in semiconductor wafers, where it excels at determining critical dimensions, thickness, and profile characteristics. However, OCD scatterometry is generally used to assess periodic gratings, and it faces significant challenges in discriminating disordered or irregular patterns, such as the artificial fingerprints created by disordered BCPs and used in our work as a PUF. Indeed, the technique averages the structural features of the analyzed area, making it difficult to devise a method for recognizing and differentiating the various patterns produced in our work and distinguishing the overall pattern, local defects and minutiae. A clear example is reported in ref¹ in which the authors proposed the use of OCD for the assignment of a grating qualification score describing the level of alignment of BCP to chemically defined guiding structures. The reported results demonstrated that OCD measurements performed in different disordered areas of the same wafer yield the same result in terms of qualification score.

Additionally, to achieve a unique and usable value for a PUF, one would need to resolve the "Non-uniqueness of solutions for inverse light scattering problem".² This issue arises because the process of determining the properties of a structure from its light scattering pattern can yield multiple possible solutions that fit the observed data, making it challenging to uniquely identify the correct physical structure.

On the other hand, we agree with the reviewer that this technique can be exploited for evaluating structural parameters such as line-edge roughness and thickness of the nanostructures.³

As detailed in R2 and R9 of the previous point-by-point response to the Reviewers, affordable tabletop SEM instruments are now available on the market, offering sufficient resolution to detect and analyze fingerprint patterns. Furthermore, recent lab-based SEM offer advanced capabilities, such as handling large objects (i.e. diameter 300 mm, height 130 mm weight 8 kg) and multi-sample imaging (scanning areas of 100 cm x 100 cm with nanometric resolution). Additionally, modern AFM/SEM instruments can now perform both types of measurements simultaneously, enabling correlative analysis of topography and morphology for more comprehensive fingerprint evaluation.^{4,5} Some of these tools can be installed in pre-existing SEM microscopes, thus enhancing the utility of SEM-based techniques in fingerprint-based PUF identification (refer to previous point-by-point response for details).

Changes in the manuscript: According to the above discussion, we added in the revised manuscript (page 22) the possibility of using optical scatterometry for evaluating morphological parameters that can be used to further check the genuinity of the physical PUF.

C2. The response says that the nanopattern can be left exposed to be accessible for imaging. Doing that for a chip is quite involved. For a chip, it means that the package would have to be partially removed to open access to the pattern. That would break the hermetic seal of the packaging and risk environmental exposure to the chip. It also requires the pattern to be made on the chip at the back end of line on top of the interconnect wiring (or a window needs to be etched into the interconnect layers to reveal a deeper layer pattern).

R2. The microchip example is just one of those we presented in our work. In our view, the potential applications of PUFs are vast, supported by the advanced maturity of nanofabrication techniques based on BCPs. These techniques allow for the creation of nanostructured patterns on both conventional substrates, such as wafers used in semiconductor device fabrication,⁶ as well as non-semiconductor applications,⁷ and even on less conventional flexible substrates.⁸ In any case, we agree with the reviewer that the use of this technology on chips would require additional steps during the fabrication process. Moreover, a key advantage of BCPs is that fabrication procedures used to induce self-assembly are fully compatible with the high-volume manufacturing processes typical of the CMOS device production.⁶

Changes in the manuscript: We further clarified in the revised text (page 24) that the chips represent only an example of applications, and that potential applications are vast thanks to the high versatility and advanced maturity of the block copolymer lithography.

C3. Additionally the nanopattern requires an additional lithography step to pattern the nanopattern to the 2-4 um square. The target has to be small and well-defined or it will be really difficult to ensure that the same spot is measured as was originally certified.

R3. It is true that the localization of the PUF pattern for authentication/identification requires an additional lithographic step to define a frame surrounding the area of interest. While the localization of the area of interest can be performed by considering markers (not necessarily a frame, but eventually also coordinates with respect to a feature of the object of interest), the realization of a frame with lithograph steps is both cost-effective and time-efficient. In our work, the frames defining the areas of interest were created using electron beam lithography (EBL) followed by metal deposition and lift-off. Due to the micrometer scale of the frames and the associated "find-me"

structures, they can be fabricated quickly using conventional methods such as optical lithography (e.g., 30-second exposure) or direct laser writer (DLW) lithography. The latter process can even be used to create patterns on curved surfaces using a laser beam.^{9,10} These types of optical lithography also allow for the use of thick and durable resists (e.g., SU-8) that can pattern rough surfaces. The compatibility between the self-assembly of BCP and lithographically defined structures created via DLW has previously been demonstrated by our group,¹¹⁻¹³ although this is beyond the scope of the current work. Additionally, the micrometer scale of the structures makes it possible to align with them using just basic optical or electron microscopy. Also, it is worth mentioning that the proposed strategy based on computer vision concepts enables robust authentication/identification of the pattern even if the end-user image is acquired with small translation, rotations and/or deformations with respect to the database one, relaxing the required alignment conditions for image acquisition.

Changes in the manuscript: According to the above discussion we added the Supplementary Note 2 at page 35 of the Supplementary Material. A sentence recalling this note has been added at page 29 of the main manuscript.

C4. The manuscript suggests that the engraved nanopattern could be put on the outside of the final product. Opportunities for that are somewhat limited since the surface where you put the nanopattern would have to be quite smooth. They discuss putting the pattern on a curved surface. That surface on a 1-2 μm length scale is relatively flat and if it was glass it would also be very smooth.

R4. In the previous point-by-point response (specifically in Figure R2), and here reported again for simplicity's sake as Figure R1, we have showed the optical profilometer map (Figure R1e) and relative height profile of the curved substrate in which the fingerprint nanopattern was engraved (Figure R1f). According to the RMS value of 104.3 nm obtained by optical profilometer measurements, this substrate cannot be considered by any means smooth as mentioned by the Reviewer (the nominal RMS of a standard polished Si wafer is < 0.5 nm). Moreover, Figure R1d shows the optical profilometer measurement of the curvature of the optical lens on a 7.3×2.5 mm^2 area in which the fingerprint nanopattern was successfully engraved over the whole surface. Additionally, we report in Figure R2 an SEM image of the engraved fingerprint nanopattern collected over a wide area of $4 \mu\text{m} \times 4 \mu\text{m}$, additionally demonstrating the successful engraving of fingerprint patterns on curved and rough surfaces.

Moreover, the possibility of creating fingerprint patterns conformal to 3D substrates made from commercial optical resins or graphene liquid crystalline fibers has also been demonstrated.¹⁴ Other research groups show stability of the self-assembled patterns after substrate stretching,¹⁵ mechanical deformation or crumpling.¹⁶

Figure R1. Fingerprint patterns engraved on a curved and rough surface. **a.** Photograph of an optical glass lens and **b.** SEM image of the engraved fingerprint pattern. **c.** Optical profilometer measurement of the curvature of the optical lens on a $7.3 \times 2.5 \text{ mm}^2$ area and **d.** relative height profile with schematic representation of the lens measured in confocal imaging mode with a 5x optical lens. **e.** Optical profilometer measurement of the roughness of the optical lens on a $42 \times 35 \text{ }\mu\text{m}^2$ area with a 20x optical lens and **f.** the relative profile.

Figure R2. Fingerprint patterns engraved on a curved and rough surface over an area of 4 μm x 4 μm . Scale bar is set to 1 μm .

Changes in the manuscript: we included as a new supplementary information the SEM image of a fingerprint pattern over a large area to further corroborate the possibility of obtaining a pattern on curved surfaces (new Supplementary Figure S2). This supplementary figure is recalled in the revised manuscript at page 7. Additionally, we included in the revised manuscript a discussion (with related references) on the possibility of realizing BCP patterns conformally on 3D structures (page 7), and a discussion (with related references) on the stability of these self-assembled nanopatterns upon stretching, mechanical deformation and crumpling (page 24).

C5. Note though that if the roughness is more than about 5-10 nm it will obscure the nanopattern. Typical metal and plastic surfaces are much rougher than that and would be unsuitable for a nanopattern PUF. For microelectronics the packaging is done at a different location than the wafer fabrication, so a PUF on the packaging is not as useful.

R5. We do not agree with the Reviewer's statement that "If the roughness is more than about 5-10 nm it will obscure the nanopattern." In fact, by appropriately selecting the thickness of the BCP film deposited via spin-coating on the substrate, it is possible to significantly reduce the impact of substrate roughness on the fingerprint pattern formed after self-assembly. The typical thicknesses of BCP films suitable for the self-assembly method using Rapid Thermal Processing, as employed in this work, exceed 30 nm. It has been demonstrated, however, that vertical orientation of BCPs can be achieved even for thicknesses up to approximately 200 nm for cylinders¹⁷ and 360 nm for lamellae.¹⁸ In some cases, roughness has been intentionally utilized to tailor the orientation¹⁹ and long-range ordering of the nanostructures.⁸

Even though it does not prevent the BCP self-assembly, roughness is certainly a potential source of disorder for the final fingerprint pattern.^{20,21} This is completely undesirable in traditional lithographic applications, such as the fabrication of regular devices for microelectronics. However, in the case of

artificial fingerprint, substrate roughness can be viewed as an additional parameter to control the level of disorder and defectivity during the self-assembly process.

Changes in the manuscript: we clarified in the revised manuscript (page 7) that the self-assembly process and resulting nanopattern can be regulated and affected by the underlying substrate roughness.

C6. The manuscript and supplemental do not appear to ever state the pitch of the block copolymer patterns and the microscope images are not consistent. Figure 1 appears to make the pitch to be about 80-100 nm (caption states that the scale bar is 400 nm and has 4 lamella in the right side and 6-7 in the left side). Figure 2 appears to make the pitch to be 33 nm (caption states the scale bar is 200 nm and has 6 lamella). Figure S2 has 7 lamella with scale bar of 200 nm (pitch of 29 nm). The AFM image in figure S24 for the cross section shows 15 lamella in 600 nm (pitch of 40 nm). They should report the pitch and check their scale bars. The methods section only lists one BCP, so one assumes all the pitches should be similar. If multiple BCPs are used, they should report that.

R6. In our work, the patterns in Fig 1f-i, Supplementary Figures S1-S7, S12, S16, S17, S20, S21, S25, S26 and the PUF devices (Figure 2,3,4, and Supplementary Figures S18 and S23) were fabricated considering BCP with $M_w = 66 \text{ kg mol}^{-1}$ that results in nanostructures with a pitch of $38.2 \pm 0.3 \text{ nm}$ (this value represents the weighted mean of the pitch value, and its error calculated by analyzing the first peaks of the fast Fourier transform over 200 images). Note that this value is consistent with one previously reported literature for BCP with the same molecular weight.^{12,13} Instead, in Figure 1d and e we reported a BCP with different molecular weight ($M_w = 42 \text{ kg mol}^{-1}$, $f_{PS} = 0.50$ and $PDI = 1.07$), associated with a pitch of $\sim 27 \text{ nm}$, as an example of typical defects in lamellar BCPs.

Changes in the manuscript: The Methods section was modified in order to clarify molecular weights exploited during the fabrication process, adding also the corresponding pitch of the nanopatters (page 26). The scale bars in Figure 1 have been corrected and the caption modified accordingly. The scale bar value of the correlation maps in Figure 2c has been added in the caption. The scale bar in Figure S3 and the relative caption have been corrected. The scale bar in Figure S12 has been corrected.

C7. Their analysis method is based on defects and fingerprint analysis. This is looking at features on the 10+ nm scale (assuming a BCP period of 30 nm). They state that one could include the nm-scale roughness of the lines in the algorithm to increase the security and entropy of the pattern. I would be very concerned with SEM repeatability if you included the line roughness in the algorithm. There is considerably work in the semiconductor industry on very expensive fab CD-SEMS to get consistent line roughness values. It is strongly dependent on the imaging conditions and how you process the data and focused on getting statistical information on roughness across a large image with lots of lines. With that approach you lose the unique information from the spatial distribution of roughness that would be required to use it towards the PUF. Repeatedly measuring the spatial fluctuations and distribution of roughness across PUF pattern would be very difficult.

R7. We agree that the measurement of LER and LWR, especially in disordered patterns such as the fingerprint-like structures formed by BCP, presents significant challenges. These challenges are well-documented in the microelectronics industry, where such parameters are typically assessed on regular gratings rather than disordered patterns, similar to the challenges encountered when measuring geometric parameters using OCD. In light of these difficulties, we deliberately chose not to incorporate LER and LWR in the current phase of our matching algorithm. While our defect recognition software, ADAblock,²² does have the capability to calculate these parameters even on

disordered patterns, we have not yet utilized this feature for the PUF analysis. However, this method could be valuable for evaluating roughness increases, typically associated with processes like imprint lithography.

We also agree with the Reviewer about the concern regarding SEM repeatability, which is indeed crucial when considering the inclusion of LER and LWR in the algorithm. Recent advancements in LER measurement techniques, however, offer promising solutions. Notably, there have been significant technical advances in off-line LER measurement that leverage common SEM tools, excluding the need for expensive online CD-SEMs, by employing sophisticated graphic recognition algorithms.²³ These developments suggest that LER and LWR could become reliable parameters for enhancing the uniqueness of fingerprint patterns in the future.

Additionally, recent studies also demonstrated that LER and LWR can be engineered during the chemical synthesis of BCP, expanding the set of variables that can be explored to enhance the unclonability of PUF.^{24,25}

Changes in the manuscript: According to the above discussion we revised the text at page 22, providing also new references to relevant works. The reference list was revised accordingly.

C8. The manuscript states that AFM could be used for looking at line roughness and the vertical dimensions of the pattern. AFM is pretty poor at nm-scale edge roughness due to tip shape effects. It would be very strongly dependent on the current tip shape and prone to artifacts. The pattern would also have to be relatively shallow for the tip to be able to fit into the ~15 nm wide trenches between the lines and reach the bottom. Most of the line shape in figure s24 is due to tip artifacts and not due to the actual shape of the engraved lines. It wouldn't be very useful for validation of a nanoscale PUF.

R8. At page 23 of the revised manuscript, we indicate the AFM as a reliable method for assessing the fingerprint pattern, comparable to SEM, but with more relaxed working conditions. For this purpose, it is not necessary for the tip to fully penetrate through the entire lamellar pattern. Furthermore, AFM is capable of providing complementary information on the nanopattern with respect to SEM analysis. This is because the measurand of the two techniques is different when aiming at the same physical quantity due to their distinct measurement principles.

At page 22 of the text, we also point out that additional morphological properties, such as LER, LWR, and 3D morphology and z-profile extracted using SPM methods, can be explored to further check the genuinity of BCP-based physical PUFs. The measurement of LER and LWR is technically feasible, and the first CD-AFM tools were developed by NIST and SEMATECH in the early 2000s. These instruments are capable of providing metrological traceability for LER measurements on nanostructures.²⁶ By utilizing ultra-sharp AFM tips that are currently commercially available, with tip diameters below 5 nm^{27,28} and commercial analysis software, offline analysis can achieve the same level of accuracy in LER quantification as online CD-SEM analysis.²⁹

Nevertheless, we agree with the Reviewer that matching the local roughness measurements of individual lamellae represents a challenge. However, in this context we believe that the global average values of LER and LWR could serve as an additional security level to assess the genuinity of the physical PUF. This is based on the observation that replication or counterfeiting processes, such as nanoimprint lithography, tend to increase the lateral roughness of the lamellae forming the fingerprint.³⁰ In this context, in line to what suggested by the Reviewer, OCS scatterometry can represent an alternative technique for assessing LER and LWR.¹

Changes in the manuscript: According to the above discussion we revised the text at page 22, providing also new references to relevant works. The reference list was revised accordingly.

C9. Nanoimprint is pretty good at replicating structures. If someone had access to the PUF pattern, you could create a replica and master stamp with ~nm-scale matching of the PUF including the roughness. The main limitation to nanoimprint resolution is the original lithography of the master template. Nanoimprint tools are also relatively inexpensive.

R9. We thank the Reviewer for the question, which gave us the opportunity to better explain the capabilities of the random fingerprints described in our work. To provide an accurate response it's essential to start with the dimensional data regarding the nanostructures formed by the BCPs, which, as noted by the reviewer, was missing in our previous submission. As we reported in R6, in our work we used two lamellar systems, with molecular weight of 66 kg mol⁻¹ and 42 kg mol⁻¹, having periodicity of 38.2 nm and 27 nm respectively. However, the lateral dimension d of the lamellae composing the fingerprint pattern is smaller and depends on the selective removal process of one of the two polymers, followed by the subsequent pattern transfer via reactive ion etching (RIE). For the previously cited BCPs with molecular weight 66 kg mol⁻¹ and 42 kg mol⁻¹, typical lateral dimensions for this type of BCP are 18.9 nm and 13.2 nm respectively.

Building on this premise, while Nanoimprint Lithography (NIL) is undeniably a powerful technique for pattern replication, it faces significant challenges in the replication of nanoscale patterns formed by BCP owning the mentioned characteristic dimensions, for the following reasons:

- **Resolution and cost:** The resolution of basic NIL tools designed for research and development with low throughput and without sophisticated alignment systems, ranges between 50 nm and 100 nm, and are priced around \$200.000 - \$500.000. These tools do not own technical features needed to correctly replicate the fingerprint patterns (i.e. resolution and alignment). More advanced systems with higher resolution (down to 20-50 nm) and better alignment capabilities could cost between \$500.000 to \$1 million. Also, in this case the resolution limit is not enough to replicate the BCP fingerprint patterns. In order to minimize (not completely avoid!) the previously described effects, advanced and industry-oriented NIL tools are required (e.g. Canon FPA-1200NZ2C, EVG 7300, SUSS MicroTec MA/BA Gen4). The estimated cost of those equipment varies between \$5 million and \$15 million. Moreover, this tool must be operated in a cleanroom environment and occupies a substantial amount of space. Although this price is significantly lower than the \$150 million price of current nanofabrication tools used in production lines (e.g. ASML EUV systems), the high cost, combined with the other challenges mentioned, limits the accessibility of NIL to few industrial state of the art laboratories.
- **Defectivity:** The formation of defects during the NIL process is a significant concern, especially when dealing with small and dense nanostructured patterns created by BCP self-assembly. Even minor defects in the mold or during the imprint process (due for example to the presence of particles or bubbles) can propagate across the replicated patterns, leading to imperfections that may affect the matching process. This is particularly critical for the application in PUFs in which the fidelity of the pattern is paramount. As mentioned before, it has also been observed a variation of LER and LWR as a typical sign of degradation of replica pattern.³⁰
- **Alignment:** Another challenge in applying NIL for the replication of fingerprint patterns is the alignment accuracy. Precise alignment between the mold and the substrate is essential to ensure that the nanostructures are replicated correctly. Misalignment can lead to distortions or inconsistencies in the pattern, which may compromise the replicated PUF functionality. Achieving the necessary alignment precision often requires sophisticated and expensive tools and techniques, adding complexity to the NIL process.

- **Replication on non-flat substrates:** Copying and transferring nanostructured patterns onto non-perfectly flat substrates presents additional difficulties. While substrate roughness and undulations do not prevent the formation of self-assembled fingerprints (as testified by the literature reported in R2, R4 and R5), it can lead to incomplete or distorted pattern transfer, reducing the effectiveness of the NIL process. Also, the presence of the frame determining the region of interest could be detrimental for the correct definition of the NIL pattern. Achieving uniform replication on such substrates often requires further optimization of the NIL process or the use of specialized techniques, which can add to the complexity and cost.
- **Replication on soft or bendable substrates:** An additional challenge arises when attempting to replicate patterns on soft or bendable substrates, which are increasingly important in flexible electronics and other emerging applications. Unlike conventional wafers, these substrates can deform during the NIL process, leading to uneven pattern transfer and defects. This specific issue occurs both in the copying and transfer phases of the pattern, consequently increasing the possibility of obtaining defects in the final fingerprint.

For these reasons, the majority of efforts that can be found in literature concerning the use of BCP and NIL are dealing with the formation of guiding structures (the so-called graphoepitaxy) that direct and enhance the ordering of self-assembled nanostructures.³¹⁻³⁴ In this process, known as “density multiplication,” the dimension of the guiding structures is way larger compared to the BCPs size. While the use of NIL in these studies is to maximize the long-range order of the BCPs, our work aims to leverage the intrinsic disorder that arises during self-assembly, without relying on physical (i.e. graphoepitaxy) or chemical (i.e. chemoepitaxy) constraints that would alter the intrinsic randomness of the system.

Reviewer #3

All of my previous comments have been addressed.

We thank the reviewer for the positive assessment of our work.

References

1. Van Look, L. *et al.* High throughput grating qualification of directed self-assembly patterns using optical metrology. *Microelectron Eng* **123**, 175–179 (2014).
2. Orji, N. G. *et al.* Metrology for the next generation of semiconductor devices. *Nat Electron* **1**, 532–547 (2018).
3. Bodermann, B., Ehret, G., Endres, J. & Wurm, M. Optical dimensional metrology at Physikalisch-Technische Bundesanstalt (PTB) on deep sub-wavelength nanostructured surfaces. *Surf Topogr* **4**, 024014 (2016).
4. <https://qd-europe.com/be/en/products/electron-microscopy/afsem-correlative-afm-and-sem/>, accessed on September 3rd 2024.
5. <https://www.nenovision.com/products/litescope-afm-in-sem>, accessed on September 3rd 2024.
6. Liu, C.-C. *et al.* Directed self-assembly of block copolymers for 7 nanometre FinFET technology and beyond. *Nat Electron* **1**, 562–569 (2018).
7. Yang, G. G. *et al.* Block Copolymer Nanopatterning for Nonsemiconductor Device Applications. *ACS Appl Mater Interfaces* **14**, 12011–12037 (2022).
8. Park, S., Lee, D. H. & Russell, T. P. Self-Assembly of Block Copolymers on Flexible Substrates. *Advanced Materials* **22**, 1882–1884 (2010).
9. Ai, J., Du, Q., Qin, Z., Liu, J. & Zeng, X. Laser direct-writing lithography equipment system for rapid and μm -precision fabrication on curved surfaces with large sag heights. *Opt Express* **26**, 20965–20974 (2018).
10. Zhang, H., Lu, Z. & Li, F. Fabrication of a curved linear grating by using a laser direct writer system. *Opt Commun* **266**, 249–252 (2006).
11. Ferrarese Lupi, F. *et al.* Hierarchical Order in Dewetted Block Copolymer Thin Films on Chemically Patterned Surfaces. *ACS Nano* **12**, 7076–7085 (2018).
12. Murataj, I. *et al.* Hyperbolic Metamaterials via Hierarchical Block Copolymer Nanostructures. *Adv Opt Mater* **9**, 1–9 (2021).
13. Ferrarese Lupi, F. *et al.* Tailored and Guided Dewetting of Block Copolymer/Homopolymer Blends. *Macromolecules* **53**, 7207–7217 (2020).
14. Yang, G. G. *et al.* Conformal 3D Nanopatterning by Block Copolymer Lithography with Vapor-Phase Deposited Neutral Adlayer. *ACS Nano* **13**, 13092–13099 (2019).
15. Cho, J. *et al.* Chiral Plasmonic Nanowaves by Tilted Assembly of Unidirectionally Aligned Block Copolymers with Buckling-Induced Microwrinkles. *ACS Nano* **15**, 17463–17471 (2021).
16. Kim, J. Y. *et al.* 3D Tailored Crumpling of Block-Copolymer Lithography on Chemically Modified Graphene. *Advanced Materials* **28**, 1591–1596 (2016).
17. Ferrarese Lupi, F. *et al.* High aspect ratio PS-b-PMMA block copolymer masks for lithographic applications. *ACS Appl Mater Interfaces* **6**, 21389–21396 (2014).
18. Wan, L., Ruiz, R., Gao, H. & Albrecht, T. R. Self-Registered Self-Assembly of Block Copolymers. *ACS Nano* **11**, 7666–7673 (2017).

19. Yager, K. G. *et al.* Disordered nanoparticle interfaces for directed self-assembly. *Soft Matter* **5**, 622–628 (2009).
20. Sivaniah, E. *et al.* Symmetric Diblock Copolymer Thin Films on Rough Substrates. Kinetics and Structure Formation in Pure Block Copolymer Thin Films. *Macromolecules* **38**, 1837–1849 (2005).
21. Zhu, Y., Aissou, K., Andelman, D. & Man, X. Orienting Cylinder-Forming Block Copolymer Thin Films: The Combined Effect of Substrate Corrugation and Its Surface Energy. *Macromolecules* **52**, 1241–1248 (2019).
22. Murphy, J. N., Harris, K. D. & Buriak, J. M. Automated Defect and Correlation Length Analysis of Block Copolymer Thin Film Nanopatterns. *PLoS One* **10**, e0133088 (2015).
23. Hu, Z. *et al.* Canny Algorithm Enabling Precise Offline Line Edge Roughness Acquisition in High-Resolution Lithography. *ACS Omega* **8**, 3992–3997 (2023).
24. Lai, H., Huang, G., Tian, X., Liu, Y. & Ji, S. Engineering the domain roughness of block copolymer in directed self-assembly. *Polymer (Guildf)* **249**, 124853 (2022).
25. Lee, K. S., Lee, J., Kwak, J., Moon, H. C. & Kim, J. K. Reduction of Line Edge Roughness of Polystyrene-*block*-Poly(methyl methacrylate) Copolymer Nanopatterns By Introducing Hydrogen Bonding at the Junction Point of Two Block Chains. *ACS Appl Mater Interfaces* **9**, 31245–31251 (2017).
26. Dixon, R. *et al.* CD-AFM reference metrology at NIST and SEMATECH. in *Metrology, Inspection, and Process Control for Microlithography XIX* (ed. Silver, R. M.) vol. 5752 324 (SPIE, 2005).
27. <https://www.opustips.com/AFM-Tip-160AC-SG>, accessed on September 3rd 2024.
28. <https://kteknano.com/?s=carbon+nanotube>, accessed on September 3rd 2024.
29. Yazgi, S. G., Ivanov, T., Holz, M., Rangelow, I. W. & Alaca, B. E. Line edge roughness metrology software. *Journal of Vacuum Science & Technology B, Nanotechnology and Microelectronics: Materials, Processing, Measurement, and Phenomena* **38**, (2020).
30. Teyssedre, H. *et al.* Repeatability of Nanoimprint Lithography Monitor Through Line Roughness Extraction. in *2020 31st Annual SEMI Advanced Semiconductor Manufacturing Conference (ASMC)* 1–4 (IEEE, 2020). doi:10.1109/ASMC49169.2020.9185377.
31. Park, S.-M. *et al.* Sub-10 nm Nanofabrication via Nanoimprint Directed Self-Assembly of Block Copolymers. *ACS Nano* **5**, 8523–8531 (2011).
32. Yang, X. *et al.* Integration of nanoimprint lithography with block copolymer directed self-assembly for fabrication of a sub-20 nm template for bit-patterned media. *Nanotechnology* **25**, 395301 (2014).
33. Simão, C. *et al.* Order quantification of hexagonal periodic arrays fabricated by *in situ* solvent-assisted nanoimprint lithography of block copolymers. *Nanotechnology* **25**, 175703 (2014).
34. Cummins, C. *et al.* Self-Assembled Nanofeatures in Complex Three-Dimensional Topographies via Nanoimprint and Block Copolymer Lithography Methods. *ACS Omega* **2**, 4417–4423 (2017).

Point-by-point response

We thank the reviewer for the fruitful discussion that gave us the possibility of further refining the quality of our manuscript. Addressing all reviewer comments, including the new experimental demonstration of the realization of BCP self-assembly on a rough rolled metal foil (as requested by the reviewer) and on a rough real object (namely, a rough nickel-plated brass clock hand), we have revised the manuscript and supplementary information (changes of this third revision are highlighted in violet). We believe that this further revised version of our manuscript including also new experimental data now complies with the high standard required for publication in *Nature Communications*.

General comments

The key aspect that we would like to clarify is the concept of unclonability. It is well known in the community that the term "unclonable" in the context of PUFs doesn't necessarily imply that cloning is impossible in principle. Instead, it means that replicating a PUF would be prohibitively expensive or impractical using current manufacturing technologies, even for the original manufacturer that knows the exact process parameters. This concept is well described in the seminal work "*Physical Unclonable Functions and Applications: A Tutorial*" in which it is stated that:

- "Further, it must be "unclonable" in the sense that the cost to engineer and manufacture another object with the same fingerprint must be prohibitively expensive or impractical using known manufacturing techniques (including by the original manufacturer).¹

As an example, the same concept is explained also in very recent studies that utilize nanometric features to create PUFs:

- "Microfabrication of similar network structures is theoretically possible, but the cost of a mass duplication of "fingerprint" labels and the intricate manufacturing technology with nanoscale accuracy eliminates the risk posed from such a laborious endeavor."²

Said that, while in principle we do not question the theoretical possibility of replicating the nanostructure pattern through NIL (note that we never questioned this possibility in our manuscript), this results to be impractical with the state-of-the art of this technology.

To the best of our knowledge, no paper in the current literature reports defectless fingerprint BCP patterns replicated using NIL. The publications mentioning NIL in conjunction with BCP employ it at the laboratory or facility level to create micrometric guiding patterns for BCP self-assembly, not for direct replication of nanoscale patterns itself.³⁻⁶ Replicating defect-free sub-20 nm patterns with NIL technology is challenging due to multiple factors, including the need for flat substrates,⁷ ultra-clean environments,⁸ and high costs associated with cleanrooms and equipment.⁹ These limitations are well-documented in literature and highlighted in the International Roadmap for Devices and Systems (IRDS).¹⁰ The difficulty in achieving defect-free replication at these scales is recognized as a significant hurdle within the scientific community, as detailed in following point-by-point responses. An additional critical point we would like to emphasize is that the Reviewer seems to overlook the second crucial step in creating our proposed PUFs, that consists in engraving the BCP pattern to the target substrate. This step can also introduce spurious defects, causing differences between the original master fingerprint pattern and the transferred one during the counterfeit process.^{11,12}

Hereafter, a detailed point-by-point response to reviewer comments is reported.

I thank the authors for their additional responses and for correcting the errors in their scale bars in the manuscript. Several of their responses are in error. These are discussed below:

Q1 The authors are in error in their comments on NIL. I'm not sure where the reported resolution of 50-100 nm for NIL came from or what is meant by resolution (discussed more below). NIL is very good at replicating structures. The biggest challenge for NIL in nanofabrication is fabrication of the original high-resolution master. In the case of their PUF, that step is already completed and just needs to be replicated. A recent review article by SV Sreenivasan (doi:10.1038/micronano.2017.75), one of the pioneers of NIL and a founder of

Molecular Imprints (now Canon), says that the resolution of NIL is < 3 nm (referencing work templating from carbon nanotubes). He also cites studies where sub-7 nm half pitch structures have been fabricated with NIL. He shows commercial examples for hard disks where companies routinely obtain sub-10 nm patterning rapidly in their production fabs. Note that the definition of resolution here is describing the half-pitch of the lines being printed, not the line edge roughness or sub-features of the lines that can be replicated. These 7-10 nm half pitch lines are less than half the width of the lines fabricated from the BCP PUF in this manuscript. Cloning with NIL will not be that difficult. Also, the resolution from a cheaper NIL system isn't going to be that worse than an expensive one. The big advantage of expensive systems is the ability to go faster with fewer defects and print much larger areas through methods like step and flash. The PUF is very small, so most of the complications of NIL will not apply. The authors also discuss challenges with NIL overlay. Overlay is unnecessary to clone the PUF. The frame around the PUF would be part of the imprint pattern and not need overlay. The alignment of the PUF with the object does not matter for authentication unless they incorporate a fiducial fabricated during a different process step. The example of sub-10 nm NIL lines for hard drives was actually one of the first use cases for BCP DSA. E-beam lithography wasn't good enough so HGST combined density multiplication DSA with an e-beam pattern to get smaller pitch structures than what they could directly print with e-beam to make their NIL masters for bit-patterned media. Now they use SADP with ebeam. Defectivity challenges for NIL are for printing large area patterns. One can be pretty defective and still get good yield on 2 μ m areas. It would also be easy for the malicious actor to make a bunch of PUFed counterfeit items and then use an SEM to reject the ones with defects.

R1. As mentioned earlier, we did not question the theoretical possibility of replicating the nanostructure pattern through NIL neither in our manuscript and previous responses. However, we do not agree with the reviewer that cloning BCP-based PUF with the state-of-the art of NIL technology will not be that difficult, as detailed below.

In our previous response we mentioned that “*while Nanoimprint Lithography (NIL) is undeniably a powerful technique for pattern replication, it faces significant challenges in the replication of nanoscale patterns formed by BCP owing the mentioned characteristic dimensions*” and we indicated five common issues related to NIL, that are quite well discussed and widely accepted in literature. For example, in ref.¹³ researchers from Toshiba Memory Corp stated that:

- *The challenges of nanoimprint lithography (NIL) are overlay, defects, throughput, template life and template patterning. [...] These challenges of NIL are sufficient to divert attention of most of the lithographers away from NIL. However, we are coming to the place to overcome these challenges by the progress of NIL system, process technology and template manufacturing technology. In this paper, we report on the latest lithography performance of NIL including half pitch (hp) 14nm patterning with single mask exposure.*

To address these challenges, researchers turn to the *Canon FPA-1200NZ2C* system (shown in Figure R1), an expensive tool designed for high-volume production on flat silicon wafers. However, this solution is not applicable in our case, as we propose applications involving also real-world objects, where the complexity and variability of surfaces make it impractical to use NIL equipment tailored for uniform, planar substrates like Si wafers.

[REDACTED]

Figure R1. Nanoimprint lithography equipment FPA-1200 NZ2C released by Canon. This equipment is designed to operate in a highly controlled cleanroom environment. Image taken from the official Canon website reported in ref.¹⁴.

Resolution.

As described in the "Resolution and Cost" subsection of the previous point-by-point response document (R9), affordable NIL tools typically achieve resolutions in the range of 50–100 nm. For instance, the device mentioned in the following link (ref.¹⁵) claims a 100 nm resolution and requires at least a class 1000 cleanroom environment. Other tools, such as those used for roll-on NIL processes,¹⁶ achieve resolutions around 200 nm. This is an order of magnitude higher than what is needed to replicate fingerprint PUFs (this is typically used for creating photonic structures).

In the previous point-by-point response document (answer **R9**), we also provided a detailed overview of the different resolution/price ranges. This clearly shows that approaching the 10 nm resolution as theoretically required for accurate PUF replication dramatically increases costs. For example, the aforementioned *Canon FPA-1200NZ2C (Figure R1)*, which potentially meets the specifications in terms of resolution needed for a malicious actor to replicate the PUF, costs approximately \$15 million. The resolution values are reported in ref.¹⁴, in which the company stated that: *“Canon's NIL technology enables patterning with a minimum linewidth of 14 nm, equivalent to the 5-nm-node required to produce most advanced logic semiconductors which are currently available. Furthermore, with further improvement of mask technology, NIL is expected to enable circuit patterning with a minimum linewidth of 10 nm, which corresponds to 2-nm-node.”*

Concerning the comment of the reviewer about the resolution of cheaper NIL systems that is not expected to be that worse than an expensive one, it is worth reminding that the IRDS in 2021 set a target of achieving 20 nm lines and spaces even for most advanced tools (see Table 1). Given the complexity of reaching this precision, it is highly improbable that this goal could be met using cheaper NIL systems. These systems lack the fine resolution and advanced capabilities necessary for such nanoscale features, which typically require high-end, expensive tools designed for cutting-edge fabrication processes (again, applicable only on flat wafers).

Based on above observations, the resolution (intended as the half-pitch of the lines being printed) required to replicate the fingerprint PUF might be achieved only by using such kind of mass production NIL tools and only on flat substrates.

[REDACTED]

Table 1. LITH-3 2021 Lithography Difficult Challenges.

Defectivity

As reported in the 2021 version of the IRDS document,¹⁰ even structures fabricated using high-throughput and high-cost NIL tools are not immune to variations in feature size, an aspect that strongly limits the possibility of cloning a BCP-based PUF with this technique. Moreover, avoiding defectivity is not only a challenge for printing large area patterns by NIL, as shown in Figure R2. Local defectivity is indeed also considered in the

review paper cited by the Reviewer,¹⁷ in which Prof. Sreenivasan lists a series of 6 typical defects that can arise during NIL replication, and that can be mitigated by a non-trivial optimization of pattern dimension, NIL conditions, and resist treatment. It is likely for this reason that part of his work was devoted to the identification of possible routes towards defect reduction.¹⁸ It is also worth noting that he also dedicated a specific section to the particle induced contamination issue. We report hereafter the text extracted from ref.¹⁷ and the related figure visually describing the defect types (Figure R2):

*“In addition to particle-induced template repeaters (discussed in section “Particle contamination and template life”), non-repeating defects can occur in the liquid phase (before UV curing) or solid phase (after UV curing). Liquid phase defects include bubbles and micro- or nanoscale voids, while solid phase defects include separation induced shear, cohesive failure of imprint materials, and feature collapse defects during or after separation. The solutions to liquid phase and solid phase defects are well understood and are summarized here. Solid phase defects are illustrated in Figures 16a–d. Defect types A, B, and C are illustrated using resist nanopillars that are the least stable resist structure possible (resist holes being the most stable). Type A defects (local feature distortion) are caused by local shear stresses; type B (cohesive failure of resist) are caused by low resist strength or high aspect ratio pillars; **type C defects (feature collapse) defects are caused by low resist modulus, high aspect ratio pillars, or very small spacing between pillars**; and type D defects (large-scale shear) are caused by macroscale shear stresses induced by uncontrolled, high-speed delamination during the separation step.”*

[REDACTED]

Figure R2. Typical NIL defects observed at local scale in replicating periodic nanoscale patterns.

Considering Resolution and Defectivity issues described in previous sections, here below we contextualize point-by-point why the state-of-the art of approaches reported in works cited by the reviewer cannot satisfy main requirements for replicating BCP-based PUFs.

1. The first article cited in the review demonstrates the reproducibility of an isolated single carbon nanotube (SWNT) (not the replication of a periodic pattern).¹⁹ As shown in Figure R3a reproduced from this work, the replication of such nanostructures results in the introduction of numerous additional defects with respect to the master template. The introduction of these spurious defects contrasts with the operational principle of PUFs based on fingerprint patterns that, for being cloned, requires a replication process that does not introduce additional defects. Note that, during replication of the nanostructure with NIL, spurious defects become apparent even in the first replication and increase with the number of subsequent replicas (Figure R3b). This effect is clearly described in the original article¹⁹ and indicated by the white arrows (as noted in the caption of original figure 4 of: “The

arrows highlight defects that appear in each of the samples.”). Additionally, in this study, the SWNTs were replicated using a transparent PDMS master, which allows UV light to pass through for a commercial photocurable polyurethane resin (i.e., Norland Optical Adhesives). The resulting lithographic template will be useless for the successive pattern transfer step into the target substrate, thus contradicting the Reviewer’s claims in Q2.

[REDACTED]

Figure R3. (a) AFM micrographs representing the SWNT morphology after replication. An increase of the overall roughness is reported. (b) Defect formation in the replica patterns after 1, 2 and 3 NIL replication processes. The white arrows show the formation of discontinuities in the SWNT replica. Adapted from ref.¹⁹.

In this context, we recently utilized similar resins (Norland Optics UV-curable Adhesives) in the realization of hyperbolic metal/dielectric metamaterials by BCP pattern replication.²⁰ Although the used fabrication procedure can not be considered as a classical NIL procedure, our experience shows that while the overall structure remains unchanged (periodic lamellar pattern), the defectivity of the resulting pattern is considerably increased with respect to the pristine template. In particular, in Figure R4a it is possible to note the formation of micro-defects due to the lack of mechanical stability of the mold. On the other hand, in Figure R4b it is evident the presence of typical nano-defects formed consisting in interruptions of linear lamellar pattern, (similar to those reported in Figure R3b). Even a small transfer of these defects will be detrimental for the extraction process of the artificial fingerprint nanopattern from micrographs described in our manuscript.

Figure R4. Formation of (a) micro-scale and (b) nano-scale defects in replicas of nanopatterns by Norland Optics resins.

2. The second work cited in the Prof. Sreenivasan review⁸ employs a process that is more plausible to be compatible with the replication of fingerprint patterns. In this case, a UV-NIL resist is used, making it applicable only with transparent molds (Figure R5a). The replicated structures are isolated and non-periodic, with sizes of approximately 30 nm (Figure R5b), while the minimum size stated in the paper refers to “*the smallest feature size on the mold of 5 nm was reproduced in the resist without tearing the polymer between the contacts.*” This indeed corresponds to the minimum distance between two Au contacts shown in Figure R5b, which, however, differs from the original mold size that is greater than 10 nm. It is therefore clear that there are dimensional discrepancies between the original pattern and the transferred pattern. Figure R5c shows a replicated 14 nm pitch pattern. The asymmetry of the resultant pattern in the bottom part of the SEM image, where one of the periodic lines is missing, indicates an issue with the pattern transfer process. Better results are obtained for 30-45 nm pitch lines. In this regard, the article clarifies that the use of NIL does not guarantee a perfect replication of the master. As stated by the authors, “*The yields of the gold contacts exceeded 90% provided dust did not contaminate the process.*” Thus, even in a dust-free environment (which is unlikely if the fingerprint pattern is engraved on a real object rather than an ideal object made in a clean room), this demonstrates that the presence of defects is quite common in NIL replicated patterns. All these observations align with what we stated in our previous response R9 in the subsection “Defectivity.” Finally, in the last section of the cited paper the authors demonstrated the reproducibility of NIL process over a 4 in. silicon wafer (Figure R5d). To this goal they patterned structures with 200 nm periodicity, one order of magnitude larger than what is observed for BCP fingerprint patterns.

All the above described aspects show that the approach reported in this work cannot satisfy requirements needed for cloning BCP-based PUF.

[REDACTED]

Figure R5. (a) NIL process used in the paper. (b) Replication of 30 nm metal contacts with original distance of 10 nm. (c) NIL on 14, 30 and 45 nm pitch lines. (d) Demonstration of NIL process over large scale with 200 nm pitch linear gratings. Adapted from ref.⁸.

3. The Reviewer also cites the 7 nm half-pitch patterns created by Hitachi Global Storage Technologies (HGST) using Jet and Flash Imprint Lithography. As shown in Figure R6a, taken from ref.¹⁷, the periodic structures exhibit noticeable variations in feature morphology compared to the original pattern. These variations appear as changes in grayscale in the SEM image (see, for example, the structures highlighted by authors through red circles in Figure R5a). Although the final nanostructures can be still exploited as memory devices, the grayscale variations are perceived as defects after the binarization process (Figure R6b). These intensity variations amplify the discrepancies between the original master pattern, obtained using a double step processing of BCP²¹ (reported in Figure R6c), and the replicated one. In Figure R7, we present additional results from the S-FIL tool cited by the Reviewer.²² As shown in Figure R7, structures approaching 20 nm are not free from defects, which are highlighted in the picture by the red circles and arrows. Again, the highly expensive approach reported in this work is unlikely to satisfy requirements needed for cloning BCP-based PUF without the introduction of spurious defects.

[REDACTED]

Figure R6. (a) 7 nm half-pitch replica of the patterns realized by HGST. (b) Binarized image. (c) Original master template fabricated by BCP directed self-assembly. Adapted from ref.¹⁷ and ref.²¹.

[REDACTED]

Figure R7. Nanostructured patterns replicated by S-FIL. (a) 21 nm half-pitch template features after fused silica etch. (b) Imprinted patterns from the templates described in (a). The red circles and arrows indicate the nanoscale defects formed in the S-FIL template. (c) Imprint from the master template of structures with feature size of approximately 100 nm. Adapted from ref.²².

4. It is worth mentioning that more advanced tools, like those cited in the review paper¹⁷, can achieve higher resolutions while mitigating the pattern defectivity, but at the expense of increased costs and even more extreme working technical conditions. This opinion is also conveyed by researchers from HGST, who pushed NIL to the limits of its capabilities, as reported in ref.⁹:

“Nanoimprint lithography (NIL) techniques use expensive machinery that relies on high temperatures and on precise controls. The ultraflat layers must be perfectly parallel to allow for an even flash layer thickness.”

For this reason, they also explored alternative methods, such as creating molds with high-resolution polymers, but their defectiveness still remains unacceptable for replicating BCP-based PUFs.⁹

All the above mentioned observations further support that, at the state-of-the art, it is objectively extremely difficult and impracticable to create an accurate replica of a PUF fingerprint, especially while considering real life objects.

Overlay and Alignment.

In our case, the overlay could definitely have an impact on the PUF counterfeiting. If it would be part of the imprint pattern (as hypothesized by the Reviewer) the frame would be etched into the same material as the fingerprint PUF. In our case however, the frame is realized using a different material with respect to the one composing the fingerprint pattern (i.e. Ti/Au, as mentioned in our manuscript in the Method section *“Fabrication of PUF devices based on artificial fingerprints at the nanoscale”*). This distinction makes it very easy to verify possible replication actions simply using SEM with backscattered electron imaging or EDX analysis by checking if the material under the frame has been previously patterned or not. While in this work, we created the frame using conventional sputtering, other techniques used in stretchable electronics could be applied to extend this approach to various objects, not just flat wafers.^{23,24} On the other hand, performing the same process on the cloned PUF pattern would require nanometric alignment capabilities, significantly impacting both the technical feasibility and the overall cost of the counterfeiting process. In this context, it is

worth noticing that overlay represents one of NIL challenges mentioned in the IRDS for further exploitation of this technology, as reported hereafter in Table 1 extracted by the 2021 IRDS document.¹⁰

We are fully aware that HGST, now part of Western Digital (a company with \$12.3 billion in revenue), has theoretically the capability in terms of fabrication facilities and competences for possible replication of our fingerprint-like patterns. Even if such replication were attempted, the reproduction of these structures with their specific dimensions and density has only been demonstrated on ultra-flat, conventional silicon wafers.²¹ Replicating them on other types of substrates, such as those described in Supplementary Figure S1-S3, would be far more challenging and unlikely to achieve the same level of precision even with the most advanced equipment.

Changes in the manuscript: we provide additional information based on the above discussion on the difficulties of replicating BCP-based nanopatterns through other techniques, with a focus on NIL technology. For this purpose, we added a discussion as a new supplementary note (Supplementary Note 2). A sentence recalling this aspect was added at page 22 of the revised manuscript. Also, we better contextualized our claims about the concept of unclonability of PUFs, based on discussions reported in literature (page 22 of the revised manuscript). Here, appropriate references were added.

Q2. The authors are also in error on their comments on CD-AFM and the ability to measure 3D morphology. CD-AFM is done on isolated lines or on lines with large space between them (<https://doi.org/10.1117/1.JMM.15.4.044006>). The patterns in this manuscript are 38 nm pitch with nominal trench width of 19 nm. CD-AFMs for nanoscale lines use a tip with a flare on the end to prevent the tip sidewall from touching the line. This makes them wider than super sharp AFM tips (often 20-50 nm). The tip also has a horizontal oscillation to do side wall profiling. The width of the tip also smooths the measured high frequency LER. Very narrow trenches require using a very high aspect ratio probe. Work has been done using carbon nanotube tips or other HAR tips. Any of the HAR probe options introduce tip flexure and likely hit the sidewall somewhere other than the end of the tip. They are very challenging to use and get repeatable results side wall profiling with CD-AFM. Conventional AFM with a 5 nm tip diameter will show you the top structure of the 38 nm pitch fingerprint patterns convoluted with tip shape. It will struggle to reach the bottom if the depth is more than the trench width. In that case it will only touch the bottom in the middle of the trench. The sidewall shape will be dominated by the tip. The 5 nm diameter is only at the end of the tip. Beyond that is a pyramidal shape that quickly tapers at an angle shallower than the sidewall angle of the line structures. 5 nm diameter tips will also smooth the high frequency LER.

R3. We thank the reviewer for pointing out this aspect that was somehow misleading in the previous version of the manuscript. What we meant with “3D morphology” (in a probably misleading way, we agree with the reviewer) was the possibility of obtaining information on the sample not only in the x,y directions, but also in the z direction. Of course, we are aware that AFM measurements will provide information that is a convolution between the 3D morphology of the sample and the AFM tip shape. However, our message was that the use of different techniques different from SEM can provide additional information that can be exploited for further checking the genuinity of the PUF device. This includes morphological parameters obtained by AFM, including local LER, LWR, and information on the z dimension (z -profiles). Considering the recent technological advancements in scanning microscopes²⁵ and the growing interest in controlling edge roughness in BCP,²⁶⁻³⁰ it is plausible to resort to AFM for the analysis of the mentioned parameters including i) height of the nanostructure and ii) LER/LWR parameters, as detailed below.

- i. Concerning the height of the nanostructure, we are aware that AFM cannot be exploited for measuring this parameter in high aspect ratio nanostructures. However, in our PUF devices where the height of the lamellae are below 10 nm, the average z height can be evaluated through AFM. In this context, we recently published a comparative study of X-ray techniques (e.g., GISAXS, GIXRF, XRR) and lab-based methods (e.g., AFM, SEM, ellipsometry) performed on samples produced via BCP lithography.³¹ (refer to Figure R8a). This study shows perfect agreement between the height measurements of lamellar structures obtained with AFM and X-ray methods, as can be observed in Figure R8b.

[REDACTED]

Figure R8. Hybrid metrology of nanostructures fabricated by BCP lithography and Sequential Infiltration Synthesis. (a) GISAXS patterns of the nanostructures and relative top-view SEM and AFM micrographs. (b) Comparison of the infiltrated nanostructures height values obtained by AFM measurements and GIXRF-XRR modeling. Adapted from ref.³¹.

In this regard, it is useful to note that, as Metrology Institute, we are currently leading an interlaboratory study, devoted to the assessment of the geometric characteristics of BCP patterns (Versailles Project on Advanced Materials and Standards (VAMAS), Strategic Activity TWA0/Project 3, “Assessing Self-Assembling Characteristics of DiBlock Copolymers (DBC) Nanostructures”) More information can be found at the link in ref.³².

Thus, in our case, AFM can undoubtedly be used to both capture the 2D morphology of the fingerprint pattern and accurately measure the height of such nanostructures, especially with thin structures such as the ones described by the AFM micrograph in Supplementary Figure S27.

- ii. Concerning the possibility of measuring LER with AFM, in 2016 researchers from Chicago University, Argonne National Laboratory and imec published in 2016 a paper entitled “*Characterization of the shape and line-edge roughness of polymer gratings with grazing incidence small-angle X-ray scattering and atomic force microscopy*”.³³ In this paper the authors analyzed by means of GISAXS and AFM periodic polymer gratings with 50 nm lateral width as a case study to be extended at Block Copolymer Directed Self Assembly. As reported hereafter the authors found a perfect agreement in the LER assessed by AFM and GISAXS (Figure R9a).

“Since the final DSA features are significantly sensitive to the geometry of the chemical patterns, a careful characterization of this grating structure, referred to as a PS guiding stripe, is crucial for optimization of the DSA process. We compare our analysis with independent characterization methods such as AFM and find good agreement between the two techniques.” Given that the characteristic dimensions of BCP can be adjusted between 80 nm and 7 nm by properly selecting its molecular weight, it is therefore plausible to consider the use of AFM in assessing line roughness, as we mentioned in the manuscript. The main discrepancies between GISAXS and AFM measurements were found in the assessment of the 3D profile (Figure R9b), as correctly observed by the Reviewer.

[REDACTED]

Figure R9. (a) Characterization of LER in polymer gratings by AFM. (b) Assessment of polymer grating vertical shape by AFM and GISAXS. Adapted from ref.³³.

Changes in the manuscript: according to reviewer comments, we specified through the manuscript that the AFM could be used to probe information about the nanostructure in the z direction. We removed the revised manuscript using the term “3D morphology” to avoid misleading interpretation from the reader (page 22), better explaining this aspect of the revised manuscript. Also, we better clarified the possibility of using AFM techniques for measuring LER and LWR, providing appropriate reference to literature (page 22).

Q3. Their discussion on surface roughness doesn't completely make sense. What matters is the nanoscale roughness within the 2 μm area of the PUF. A curved surface could still be smooth locally on the nanoscale. It doesn't surprise me that a polished optical lens would have lots of areas locally smooth enough for BCP self assembly. What was the local surface roughness in the 4 μm region in figure R2? Nanoscale roughness will affect the ability of the BCP to form vertical lamella. You likely will end up with mixed orientation of the BCP. Thicker BCPs will probably make it worse because you would get multiple orientations in the depth of the film and the pattern transfer would be poorly defined. High frequency roughness will affect the ability to transfer the pattern to the substrate. It is possible as the authors point out that mixing of the high frequency substrate roughness with the BCP fingerprint will add to the entropy in some cases. It also will make it more difficult to cleanly image the resulting fingerprint. The 104.3 nm RMS value for the optical profilometer was over a 35x42 μm field with a method with micron resolution. The surface slopes are pretty smooth so a lot of local areas might not have a lot of nanoscale roughness. One conceivably could identify a smooth enough spot for the PUF or use some sort of local polishing method to make a smooth spot. They should try something like a rolled metal foil. The referenced paper for tall lamella is using a multistage directed self assembly process to get the tall lamella. That process wouldn't work on really rough substrates.

R3. Based on literature and on our previous works, we do not agree with the reviewer that the self-assembly and transferring process wouldn't work on rough substrates, as detailed below. Of course, we are aware that the process parameters (including polymer thickness, substrate functionalization, etc.) should be optimized depending on the peculiar roughness of the target substrate.

In this context, several studies have shown that BCP self-assembly can adapt to rough or irregular surfaces, maintaining (of course, under specific conditions) nanopatterns over large scales as required for BCP-based PUFs. We report for the benefit of the reviewer some examples of BCP self-assembled on rough surfaces at both nano-scale and micro-scale (Figure R10a-d).³⁴⁻³⁷ In Supplementary Figure S3 we reported an example of BCP self-assembly performed on a real object (the optical lens). Here, the RMS roughness calculated over

several $4 \times 4 \mu\text{m}^2$ regions is in the range of about 50-170 nm, depending on the specific area. In this context, the BCP self-assembly was observed to occur unregarding the specific local roughness.

[REDACTED]

Figure R10. Block copolymer self-assembly on rough substrates at nano-scale (a)-b and micro-scale (c)-(d). Adapted respectively from (a) ref.³⁴, (b) ref.³⁷, (c) ref.³⁵ and (d) ref.³⁶.

As requested by the reviewer, we experimentally demonstrate BCP self-assembly on a rough rolled metal foil (Figure R11). In particular, commercial Cu foil underwent the same fabrication steps for the promotion of BCPs fingerprint patterns as already reported in the methods section (“*Fabrication of fingerprint patterns by block copolymers*”) for different materials (silicon, silicon oxide, quartz, diamond).

In order to further prove the possibility of obtaining self-assembly on rough real objects, we performed the self-assembly process over a nickel-plated brass (Copper-Zinc alloy) clock hand Figure R12a. Over a large $90 \mu\text{m} \times 120 \mu\text{m}$ area the RMS roughness is 79.3 nm (Figure 12b), while at a local scale, over a $4 \times 4 \mu\text{m}^2$ area, its value is 33.2 nm (Figure R12c). SEM micrographs showing the high roughness of the substrate are reported in Figure R12 f-g, and the successful self-assembly of BCP over the rough object is reported in Figure R12h.

Figure R11. BCP self-assembly on a rolled metal foil. (a) Photograph of the Cu metal foil used as a substrate for the BCP self-assembly. (b, c) Optical profilometry topographic maps at two different scales and (d, e) corresponding height profiles traced by the red lines in (b, c), showing the high roughness of the substrate. The RMS value measured in (b) is 114.6 nm; the RMS value measured in (c) is 56.2 nm. SEM micrographs at different magnifications showing the high roughness of the substrate (f and g), and a high magnification image showing self-assembled BCP over the selected object (h). Scale bars are set to 10 μm for (f), 3 μm for (g) and 200 nm for (h).

Figure R12. BCP self-assembly on a nickel-plated clock hand. (a) Optical image of the nickel-plated brass clock hand used as a substrate for the BCP self-assembly. Scale bar is set to 4 mm. (b, c) Optical profilometry topographic maps at two different scales and (d, e) corresponding height profiles traced by the red lines in (b, c), showing the high roughness of the substrate. The RMS value measured in (b) is 79.3 nm; the RMS value measured in (c) is 33.2 nm. SEM micrographs at different magnifications showing the high roughness of the substrate (f and g), and a high magnification image showing self-assembled BCP over the selected object (h). Scale bars are set to 2 mm for (f), 2 μm for (g) and 500 nm for (h).

Concerning the referenced paper for tall lamella, previous works show that it is possible to obtain perpendicular orientation even over hundreds of nanometers by appropriately adjusting the thickness of the BCP. For example, we showed that in our previous work.³⁸ In a subsequent work, we have also demonstrated the possibility of achieving pattern transfer onto lamellar structures exceeding 100 nm in height.³⁹ The second study, conducted by three researchers from HGST, presents lamellar structures with a height of 360 nm.⁴⁰ The “multistage directed self-assembly” or more correctly “self-registered self-assembly” (SRSA) is intended to ensure that lamellar structures follow the pre-patterned substrate, resulting in highly oriented and defect-free structures. As illustrated in Figure R13, if the self-registration condition is not met, thick but disordered patterns are obtained, which is precisely what we aim to achieve in our PUF, but no alteration of the internal lamellar structure is observed. Recently researchers have also demonstrated that it is possible to induce

perpendicular orientation of thick nanostructures irrespectively on the substrate using a technique based on filtered plasma.⁴¹

[REDACTED]

Figure R13. Demonstration of the SRSA process on conventional and self-registered chemical patterns. Reproduced from ref.⁴⁰.

Concerning the influence of substrate roughness on pattern transfer, this does not represent an obstacle for our purpose on conventional substrates/objects (as shown for example by transferring the pattern on the optical lens). This is because the transfer of the pattern from the BCP mask to the underlying substrate (engraving process) relies on an anisotropic chemical or ion etching that is not limited by the substrate roughness. It is worth mentioning that the use of a rough substrate will actually increase the difficulty of replicating the pattern through NIL technology, as reported in ref.¹⁷.

Changes in the manuscript: we provide new experimental data clearly showing that BCP self-assembly can be achieved on rough surfaces. As requested by the reviewer, we experimentally demonstrate that BCP self-assembly can be performed on a rough rolled metal foil in a new supplementary figure (new Supplementary Figure S1). In addition, we provide experimental demonstration of the possibility of realizing BCP self-assembly on a rough real object, namely a nickel-plated brass clock hand (as reported in the new Supplementary Figure S2). A sentence recalling these additional experimental data has been added in the revised manuscript (page 7), we also revised the method section including fabrication details accordingly. Also, we included new references (page 26 of the manuscript) discussed above further supporting the possibility of BCP self-assembly on a wide range of rough substrates. As requested by the reviewer, we also added the RMS roughness over $4 \times 4 \mu\text{m}^2$ regions of the lens in the caption of Supplementary Figure S3, clarifying that BCP self-assembly was observed to occur unregarding the specific local roughness.

References

1. Herder, C., Yu, M.-D., Koushanfar, F. & Devadas, S. Physical Unclonable Functions and Applications: A Tutorial. *Proceedings of the IEEE* **102**, 1126–1141 (2014).
2. Sun, N. *et al.* Random fractal-enabled physical unclonable functions with dynamic AI authentication. *Nat Commun* **14**, 2185 (2023).
3. Voet, V. S. D. *et al.* Interface Segregating Fluoralkyl-Modified Polymers for High-Fidelity Block Copolymer Nanoimprint Lithography. *J Am Chem Soc* **133**, 2812–2815 (2011).
4. Park, S.-M. *et al.* Sub-10 nm Nanofabrication via Nanoimprint Directed Self-Assembly of Block Copolymers. *ACS Nano* **5**, 8523–8531 (2011).
5. Yang, X. *et al.* Integration of nanoimprint lithography with block copolymer directed self-assembly for fabrication of a sub-20 nm template for bit-patterned media. *Nanotechnology* **25**, 395301 (2014).
6. Borah, D. *et al.* Soft Graphoepitaxy for Large Area Directed Self-Assembly of Polystyrene- *block* - Poly(dimethylsiloxane) Block Copolymer on Nanopatterned POSS Substrates Fabricated by Nanoimprint Lithography. *Adv Funct Mater* **25**, 3425–3432 (2015).
7. Ji, R. *et al.* UV enhanced substrate conformal imprint lithography (UV-SCIL) technique for photonic crystals patterning in LED manufacturing. *Microelectron Eng* **87**, 963–967 (2010).
8. Austin, M. D. *et al.* Fabrication of 5nm linewidth and 14nm pitch features by nanoimprint lithography. *Appl Phys Lett* **84**, 5299–5301 (2004).
9. Williams, S. S. *et al.* High-Resolution PFPE-based Molding Techniques for Nanofabrication of High-Pattern Density, Sub-20 nm Features: A Fundamental Materials Approach. *Nano Lett* **10**, 1421–1428 (2010).
10. *INTERNATIONAL ROADMAP FOR DEVICES AND SYSTEMS 2021 UPDATE LITHOGRAPHY THE IRDS IS DEVISED AND INTENDED FOR TECHNOLOGY ASSESSMENT ONLY AND IS WITHOUT REGARD TO ANY.* (2021).
11. Dialameh, M. *et al.* Influence of block copolymer feature size on reactive ion etching pattern transfer into silicon. *Nanotechnology* **28**, 404001 (2017).
12. Frascaroli, J., Seguni, G., Spiga, S., Perego, M. & Boarino, L. Fabrication of periodic arrays of metallic nanoparticles by block copolymer templates on HfO₂ substrates. *Nanotechnology* **26**, 215301 (2015).
13. Nakasugi, T. *et al.* Half pitch 14 nm direct patterning with Nanoimprint lithography. in *2018 IEEE International Electron Devices Meeting (IEDM)* 11.7.1-11.7.4 (IEEE, 2018). doi:10.1109/IEDM.2018.8614578.
14. <https://global.canon/en/product/indtech/semicon/fpa1200nz2c.html>, accessed on September 26th 2024.
15. <https://www.kyodo-inc.co.jp/english/electronics/nanoimprint/machine-tex.html>, accessed on September 26th 2024.
16. Leitgeb, M. *et al.* Multilength Scale Patterning of Functional Layers by Roll-to-Roll Ultraviolet-Light-Assisted Nanoimprint Lithography. *ACS Nano* **10**, 4926–4941 (2016).
17. Sreenivasan, S. V. Nanoimprint lithography steppers for volume fabrication of leading-edge semiconductor integrated circuits. *Microsyst Nanoeng* **3**, 17075 (2017).
18. Ye, Z. *et al.* Defect reduction for semiconductor memory applications using jet and flash imprint lithography. *Journal of Micro/Nanolithography, MEMS, and MOEMS* **11**, 031404–1 (2012).

19. Hua, F. *et al.* Polymer Imprint Lithography with Molecular-Scale Resolution. *Nano Lett* **4**, 2467–2471 (2004).
20. Murataj, I. *et al.* Hyperbolic Metamaterials via Hierarchical Block Copolymer Nanostructures. *Adv Opt Mater* **9**, 1–9 (2021).
21. Ruiz, R., Dobisz, E. & Albrecht, T. R. Rectangular patterns using block copolymer directed assembly for high bit aspect ratio patterned media. *ACS Nano* **5**, 79–84 (2011).
22. <https://www.spie.org/news/0823-imprint-lithography-template-fabrication-for-emerging-market-applications>, accessed on September 26th 2024.
23. Zumeit, A., Dahiya, A. S., Christou, A., Shakthivel, D. & Dahiya, R. Direct roll transfer printed silicon nanoribbon arrays based high-performance flexible electronics. *npj Flexible Electronics* **5**, 18 (2021).
24. Heo, S. *et al.* Instant, multiscale dry transfer printing by atomic diffusion control at heterogeneous interfaces. *Sci Adv* **7**, (2021).
25. Hu, Z. *et al.* Canny Algorithm Enabling Precise Offline Line Edge Roughness Acquisition in High-Resolution Lithography. *ACS Omega* **8**, 3992–3997 (2023).
26. Murphy, J. G., Raybin, J. G. & Sibener, S. J. Mapping the Dynamics of Fluctuations and Defects in Confined Block Copolymer Films with High-Speed Atomic Force Microscopy. *Macromolecules* **57**, 7270–7279 (2024).
27. Lai, H., Huang, G., Tian, X., Liu, Y. & Ji, S. Engineering the domain roughness of block copolymer in directed self-assembly. *Polymer (Guildf)* **249**, 124853 (2022).
28. Patrone, P. N. & Gallatin, G. M. Modeling Line Edge Roughness in Templated, Lamellar Block Copolymer Systems. *Macromolecules* **45**, 9507–9516 (2012).
29. Stoykovich, M. P. *et al.* Remediation of Line Edge Roughness in Chemical Nanopatterns by the Directed Assembly of Overlying Block Copolymer Films. *Macromolecules* **43**, 2334–2342 (2010).
30. Sunday, D. F. *et al.* Influence of Additives on the Interfacial Width and Line Edge Roughness in Block Copolymer Lithography. *Chemistry of Materials* **32**, 2399–2407 (2020).
31. Murataj, I. *et al.* Hybrid Metrology for Nanostructured Optical Metasurfaces. *ACS Applied Materials and Interfaces* Preprint at <https://doi.org/10.1021/acsami.3c13923> (2023).
32. http://www.vamas.org/twa0/documents/2024_vamas_twa0_p3_DBC.pdf.
33. Suh, H. S. *et al.* Characterization of the shape and line-edge roughness of polymer gratings with grazing incidence small-angle X-ray scattering and atomic force microscopy. *J Appl Crystallogr* **49**, 823–834 (2016).
34. Sivaniah, E. *et al.* Symmetric Diblock Copolymer Thin Films on Rough Substrates. Kinetics and Structure Formation in Pure Block Copolymer Thin Films. *Macromolecules* **38**, 1837–1849 (2005).
35. Yang, G. G. *et al.* Conformal 3D Nanopatterning by Block Copolymer Lithography with Vapor-Phase Deposited Neutral Adlayer. *ACS Nano* **13**, 13092–13099 (2019).
36. Park, S., Lee, D. H. & Russell, T. P. Self-Assembly of Block Copolymers on Flexible Substrates. *Advanced Materials* **22**, 1882–1884 (2010).
37. Yager, K. G. *et al.* Disordered nanoparticle interfaces for directed self-assembly. *Soft Matter* **5**, 622–628 (2009).
38. Ferrarese Lupi, F. *et al.* High aspect ratio PS-b-PMMA block copolymer masks for lithographic applications. *ACS Appl Mater Interfaces* **6**, 21389–21396 (2014).

39. Ferrarese Lupi, F. *et al.* Tailored and Guided Dewetting of Block Copolymer/Homopolymer Blends. *Macromolecules* **53**, 7207–7217 (2020).
40. Wan, L., Ruiz, R., Gao, H. & Albrecht, T. R. Self-Registered Self-Assembly of Block Copolymers. *ACS Nano* **11**, 7666–7673 (2017).
41. Oh, J. *et al.* Universal perpendicular orientation of block copolymer microdomains using a filtered plasma. *Nat Commun* **10**, 2912 (2019).

Point-by-point response

We thank the reviewer for the fruitful discussion that helped in further improving the quality of our work and for having recognized our efforts in improving the manuscript. We believe that this further revised version of our manuscript, including editorial revisions (changes of this last round of review are highlighted in red), now complies with the high standard required for publication in *Nature Communications*.

Reviewer #2

The authors have gone above and beyond in their reply and revisions. I didn't intend to make them do that much extra work.

I'm glad to see they have corrected the section on 3D imaging with AFM. The revisions are sufficient.

The NIL discussion was very thorough. I do still think it would not be that difficult for someone skilled in NIL to make 20 copies and then look at them with an SEM to find one with sufficient matching to the original pattern. The cloning though is irrelevant to the definition of PUF that they described where "unclonable" just means difficult to clone. This section is ok now.

We thank the reviewer for appreciating the improvement of our work.

C1. I do still have problems with the section on fingerprints on rough surfaces. PS-PMMA is somewhat neutral in chi with air. The lamella in thick films often will template from the air-BCP interface. The BCP film is also not perfectly conformal and will likely be smoother for high frequencies than the underlying surface. The SEM is looking at the top surface. Seeing fingerprints on the top surface says nothing about the structure below the surface. There is no driving force to make lamella go all the way from the surface to the substrate with a perfect vertical orientation. Thick films could have multiple orientations under the surface or tilted lamella. Both of these cases will cause big challenges with transferring the fingerprint pattern to the substrate. Tilted lamella probably won't print. It isn't necessary for them to try the etch to transfer the BCP fingerprints to a rough surface. If it is easy to do, that would improve the manuscript. If they don't do the etch transfer, then they need to add a comment about how seeing the fingerprint on the BCP surface does not mean that it will actually be able to be transferred. Once they correct that section the manuscript will be ready to publish.

R1. Concerning driving forces to make lamella, we would like to point out that also the bottom surface is neutral because of the presence of the random copolymer, and by properly tuning the thickness of the block-copolymer film it is possible to make the lamella go all the way (as discussed in previous works [refs]). Despite we demonstrated a successful engraving process in a rough (and curved) quartz substrate (Supplementary Figure S3, we agree with the reviewer that the process to transfer of the BCP fingerprint to a rough surface needs to be optimized depending on the peculiar target substrate. In these circumstances, a successful engraving process relies on the optimization of the pattern transfer process depending on both the target substrate material and roughness.

Changes in the manuscript: according to the reviewer comment, we have updated the manuscript clarifying that a successful engraving process relies on the optimization of the pattern transfer process depending on both the target substrate material and roughness (page 7 of the revised manuscript).